# Plastic associated endocrine disruptors reduce Nicastrin protein and potentiate inflammation in hidradenitis suppurativa skin disease

Kaitlin L. Williams [1], Beita Badiei [1], James Reilly[1], William Andrews[2], Hana B. Minsky [1], Nina Rossa Haddad[1], Eddie G. Martinez Pena [1], Mengqi Sun[1], Sam Lee [1], Ang Li[1], Leigh Curvin-Aquilla[1], Arieana Y. Johnson [1], Aiden Willis[1], Charles S. Kirby[1], Amy van Ee [1], Yingchao Xue[1], Carrie A. Cox[3], Shanmuga Priya Rajagopalan [3], Sewon Kang[1], Kurunthachalam Kannan[4], Julie Caffrey[3], Nathan K. Archer [1], Maureen Kane [2] & Luis A. Garza [1,5,6] ✉

Hidradenitis Suppurativa (HS) is an inflammatory skin disorder with limited treatments and unclear etiology. While monogenic HS is linked to gamma secretase mutations, particularly in the NCSTN subunit, the pathogenesis of the more common sporadic form remains uncertain, though associated with risk factors such as diets high in ultra-processed foods. Consistent with the clinical overlap between sporadic and monogenic HS, we find loss of NCSTN protein in sporadic HS fibroblasts. We hypothesize the rising incidence of sporadic HS and its hormonal associations implicate endocrine-disrupting chemicals, especially plastic-associated EDCs (p-EDs) common in UPFs. We detect elevated p-ED adducts in HS skin, persisting in ex vivo cultured fibroblasts. At nanomolar concentrations, p-EDs inhibits NCSTN and primes fibroblasts for inflammation, mimicking NCSTN knockdown. These findings suggest p-ED exposure contributes to HS pathogenesis, highlighting the need to address environmental exposures in HS and other gamma secretase-related diseases.

Hidradenitis suppurativa (HS) is a chronic inflammatory skin condition often characterized by nodules, cysts, and tunneling sinus tracts[1]. Incidence of HS varies by study, anywhere from 0.1% to 2% of the general population, and is increasing over time[2–4]. Despite its common occurrence, the only FDA-approved therapies for HS are antibodies against TNFα (Adalimumab) and IL-17A (Secukinumab)[5,6]. Response rates can be as low as 40%, underscoring the necessity of further investigation and new therapies[7,8]. Female sex, obesity, and metabolic syndrome have the strongest associations with HS, alongside smoking[2,9–11]. In addition to metabolic syndrome, there are other hormonal associations with HS—the average age of onset is between 21 and 24, and prepubertal HS is rare[2,12–15]. Additionally, 78% of adult females with HS report flares associated with their menstrual cycle[16,17]. Another association in HS is ultra-processed foods (UPFs) consumption; Non-Hispanic Black populations consume the most UPFs and have the highest HS incidence, while Hispanic populations consume

[1]Department of Dermatology, Johns Hopkins University School of Medicine, Baltimore, MD, USA. [2]Department of Pharmaceutical Sciences, University of Maryland School of Pharmacy, Baltimore, MD, USA. [3]Department of Plastic and Reconstructive surgery, Johns Hopkins University School of Medicine, Baltimore, MD, USA. [4]Wadsworth Center, New York State Department of Health, Albany, NY, USA. [5]Department of Oncology, Johns Hopkins University School of Medicine, Baltimore, MD, USA. [6]Department of Cell Biology, Johns Hopkins University School of Medicine, Baltimore, MD, USA. ✉e-mail: LAG@jhmi.edu

the least UPFs and have the lowest HS incidence, despite similar rates of obesity[18–22]. Overall, the association of HS with UPF consumption and endocrine disorders suggests novel avenues of investigation.

Many descriptive studies of HS characterize the cellular and molecular milieu. scRNAseq studies demonstrate unique adaptive immunity, fibrotic or inflammatory fibroblasts, and myeloid features in late-stage HS[23–25]. Metabolomics of HS highlights associations with metabolic syndrome[9,26,27]. Other studies detect changes in lipid or cholesterol metabolism[28,29], and dysregulation of tryptophan catabolism, affecting skin-microbiota interactions[30]. Proteomic analysis of HS skin has found the inflammatory signature of HS to be more broad and intense than psoriasis skin, and that nonlesional skin also shows hallmarks of inflammation, suggesting HS is a systemic disease not limited to skin lesions[31]. In confirmation, elevated neutrophil-related cytokines exist in the serum of patients whose skin biopsies contain high neutrophil infiltration[32]. Current literature supports HS as a highly inflammatory disease with dysregulated metabolism. However, given the extreme burden of disease in many studied patients, untangling causative factors from secondary features in HS has been challenging.

Perhaps the clearest insights into HS pathogenesis are the estimated 1–5% of patients with autosomal dominant mutations, usually in subunits of the gamma secretase (GS) protein complex, with the majority of those in Nicastrin (NCSTN)[14,33–35]. GS is a membrane-bound protease with promiscuous activity; the only currently known requirement for a GS ligand appears to be a single transmembrane pass, and there are more than 90 candidate substrates[36–38]. NCSTN is thought to act as the guardian of the complex, affecting maturation and stability as well as controlling ligand access to proteolytic cleavage at the active site[39–44]. These ligands include Notch receptors, amyloid precursor protein (APP), TNFR, E-Cadherin, IL-1R1, and IGF-1R, among others[36]. GS mutations are also a mechanism for early-onset familiar Alzheimer's disease, the focus of the bulk of research on GS biology[45–49]. More work defining GS biology will likely illuminate early HS pathogenesis.

UPF consumption and endocrine function in HS deserve further investigation. Given the similarity between monogenic and sporadic HS, their effects on GS biology will be critical to understand. The endocrine-disrupting chemicals (EDCs) abundant in UPFs include the plastic-associated phthalates and bisphenols (hereafter plastic-associated endocrine disruptors; p-EDs), common in food packaging[50–54]. Consistent with the timeframe of rising HS incidence, p-EDs have similarly increased in abundance in human tissue over the last 30 years[50,51,53,55] with measurable concentrations in both urine and serum in most humans[51,56]. For example, bisphenol A (BPA) was detected in 80–100% of study populations in multiple tissue and fluid types[3,4,50,53–55,57]. Further, the storage of blood in bags containing phthalates promotes IL-8 secretion in blood cells and epithelial cells, suggesting an inflammatory role of p-EDs[58,59]. Perhaps most intriguing of all, BPA is reported to bind and impact GS[60]. In summary, considerable evidence links p-EDs, GS, and inflammation, suggesting the need for studies in HS—but with careful controls given the ubiquity of p-EDs in the global population.

In this article, we identify a selective loss of NCSTN in dermal fibroblasts of sporadic HS patients and show how NCSTN loss potentiates inflammation in fibroblasts. Through mass spectrometry imaging (MSI), we also discover that p-EDs are significantly enriched in HS tissue and in ex vivo cultured HS fibroblasts compared to site-matched controls. Finally, we demonstrate how p-EDs induce NCSTN protein loss to prime inflammatory signaling at nanomolar concentrations. This work links the pathogenesis of sporadic HS to inherited HS, where p-EDs are present and degrade NCSTN protein to functionally recapitulate the DNA mutations of NCSTN in inherited HS.

# Results

## Dermal fibroblasts in HS express low NCSTN in tissue histology and ex vivo

HS is a debilitating dermatologic condition that typically effects intertriginous zones of the body (Fig. 1a). It is also increasing in incidence over time when compared to other dermatologic conditions, such as Atopic Dermatitis (Fig. 1b). Given the low efficacy of medical interventions for HS, excisional surgery is a common and effective modality to treat the mutilating lesions of late stage HS, especially of the axillae. We processed excisional surgery tissue from sporadic HS patients for multiple applications, including sectioning tissue for immunofluorescence (IF) and extracting cells for in vitro studies (Supplementary Fig. 1a). Demographics and characterization of patients are listed in Table 1; a total of 12 HS patients and 9 normal axillary samples were obtained for analysis. HS patient tissue was stained for NCSTN, PDGFRα (a fibroblast marker[61,62]), and DAPI (Fig. 1c). Representative H&E histology of patient excisional samples is presented in Supplementary Fig. 2. When compared to normal axillary control tissue (CNTRL) and averaged per sample, NCSTN expression is significantly lower in HS dermal fibroblasts, but not different in HS keratinocytes (Fig. 1d). This relationship holds when all patient fibroblasts and keratinocytes are pooled (Supplementary Fig. 1b). Consistent with this finding, HS fibroblasts have less NCSTN than HS keratinocytes, but CNTRL fibroblasts have comparable NCSTN levels to CNTRL keratinocytes (Supplementary Fig. 1c). The full distribution of NCSTN in sample keratinocytes and fibroblasts is depicted in Supplementary Fig. 1d and 1e, respectively. We hypothesize that NCSTN loss in fibroblasts of HS patients is likely specific to sites of HS and found that a tissue sample from the periphery of an excision has significantly more NCSTN in dermal fibroblasts than a central section of the excision (Supplementary Fig. 1f). In summary, sporadic HS patients have selective loss of NCTN protein expression in dermal fibroblasts by histology.

To verify this finding, we sub-cultured primary fibroblasts from HS surgical excisions. Extracted fibroblasts in vitro maintain this NCSTN loss as detected by IF (Fig. 1e, quantified in 1f) (pooled HS vs CNTRL samples and full distribution Supplementary Fig. 1g, h) and by western blot (representative Supplementary Fig. 1i), both in neutral conditions (Fig. 1g) and under conditions of inflammatory stress with TNFα (Fig. 1h). These results suggest a persistent cellular memory for loss of NCSTN protein and confirm that dermal fibroblasts of patients with sporadic HS have selective loss of NCSTN protein expression—the most mutated gene in the monogenic form of HS.

## Fibroblasts are the strongest immune signalers in HS by scRNAseq meta-analysis

Given the selective loss of NCSTN in HS fibroblasts, we queried the general activity of fibroblasts in HS. We characterized the role of fibroblasts in HS using a meta-analysis of two scRNAseq studies (GSE154775[25] and GSE175990[23]). We combined all samples and analyzed them using R packages Seurat and CellChat. Cells clustered into 18 distinct cell groups (Fig. 2a). We determined their identities manually using the top five gene expressions from each cluster (Supplementary Fig. 3a). We next analyzed general signaling between all cell clusters (Supplementary Fig. 3b), and specifically from fibroblasts (with or without collagen; Supplementary Fig. 3c, d, respectively). HS is known to be associated with changes in many inflammatory genes, particularly in the CXCL- and CCL- cytokine families. For example, CXCL13, a B-cell chemotactant, has received increasing attention in HS and is upregulated 16-fold in scRNAseq analysis of HS skin. Similarly, CXCL1, a neutrophil chemotactant, is upregulated 2.8-fold in the same data set[25]. CCL2, 3, 4, and 5 have also been found to be elevated in HS skin, among others[63]. Because HS is so highly inflammatory, with increased expression of many cytokines, we created a module

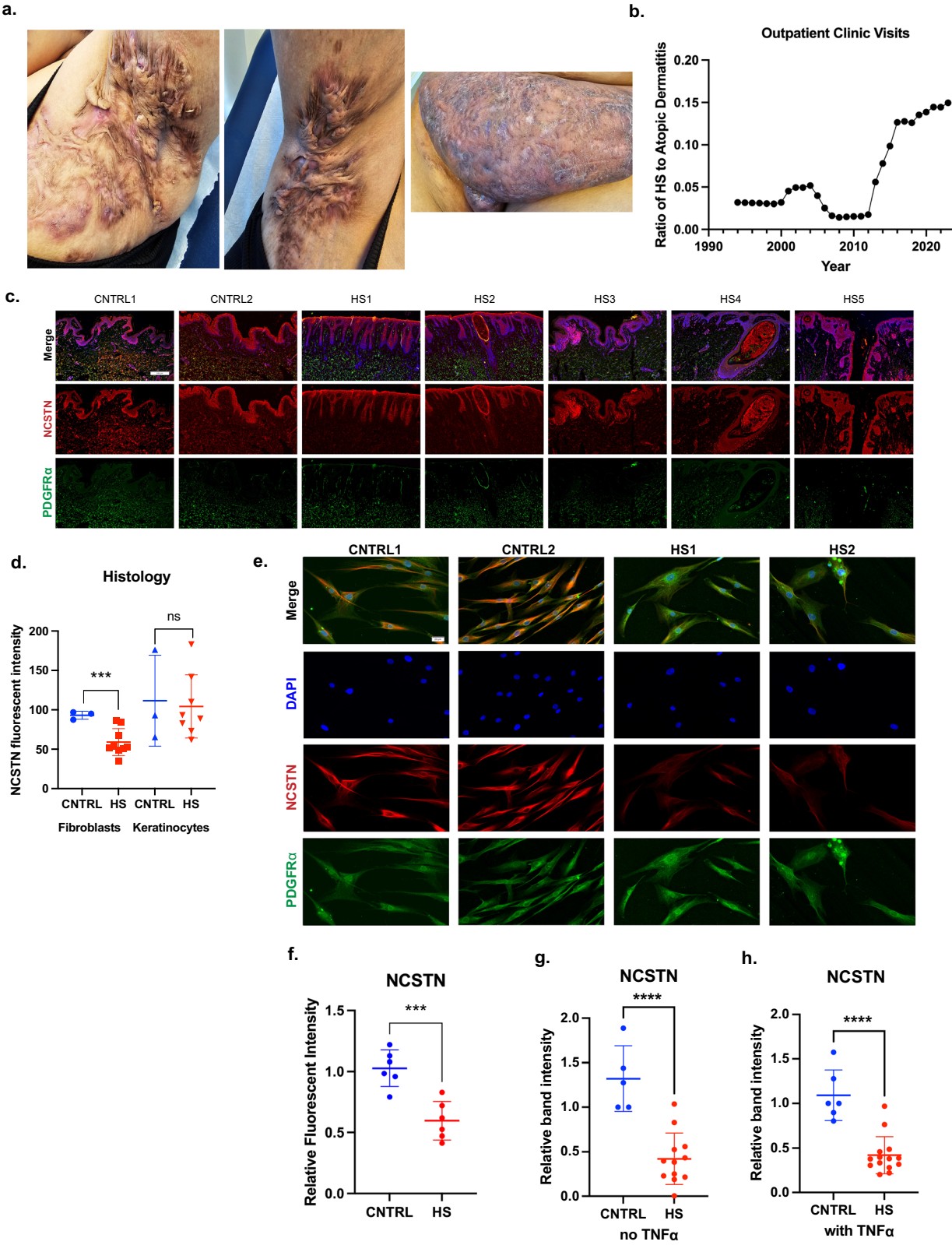

expression score that combined all CXCL and CCL genes to get a broad sense of cytokine and inflammatory signaling. Using this module score, fibroblasts are found to be the strongest signalers. This is appreciable both by a clustered FeaturePlot (Fig. 2b) and by chord diagram (Fig. 2c). Compared to other cell types, including immune cell populations and keratinocytes, fibroblasts have the highest outgoing interaction: incoming interaction ratio (Fig. 2d). As an example, there is

a 4.869 log fold change (adjP = 2.77E-221) in CXCL13 expression in HS fibroblasts compared to a healthy control (HC), which is 9-fold higher than next most abundant CXCL13 producer, T cells. Consistent with a model where fibroblasts are originators of immune signals rather than perpetuators, fibroblasts are the strongest senders of CXCL and CCL signals and are also strong influencers of these signals (Fig. 2e). Interestingly, fibroblasts subcluster into three distinct populations

**Fig. 1 | Dermal fibroblasts in hidradenitis suppurativa (HS) express low nicastrin (NCSTN) protein in tissue histology and in vitro. a** Hurley Stage III HS in a patient, concentrated on the axillae and groin. **b** The incidence of HS has grown over time when compared to another dermatologic condition, atopic dermatitis, suggesting an environmental change or increased diagnostic ability. Dermal fibroblasts, but not epidermal keratinocytes, express less NCSTN (red) in HS skin compared to healthy axillary tissue controls (CNTRL) as shown in representative tissue immunofluorescence (IF; **c** or in quantification (**d**) (keratinocytes; $p = 0.8538$, fibroblasts $p = 0.0007$; $n = 8$ patients and 3 healthy controls replicated in 3

experiments). Extracted dermal fibroblasts from HS patients maintain reduced NCSTN expression when cultured in vitro as shown in representative immunofluorescence images (**e**) or quantification (**f**) ($p = .0007$; $n = 6$ patients and 6 healthy controls replicated over 2 experiments). **g–h** In vitro HS fibroblasts cultured with and without TNFα express less NCSTN in western blot quantification ($p < .0001$, $n = 12$ HS patients and 3 CNTRL replicated over 2 experiments; (**g**, **h**). Quantification values are expressed as mean ± SEM, $p$ values based on one-way ANOVA tests or two-sided Student's $t$ tests. Source data are provided as a Source Data file.

## Table 1 | Demographic information of study participants

| PUB ID | Site of surgery | Sex | Age | Race | BMI category | Hurley stage |
|--------|----------------|-----|-----|------|--------------|--------------|
| HS1 | left axilla | F | 40–50 | White | Obesity class II | 3 |
| HS2 | lower abdomen, mons pubis, bilateral groin | F | 30–40 | Asian | Obesity class II | 3 |
| HS3 | bilateral buttocks | F | 60–70 | African American | Obesity class III | 3 |
| HS4 | groin, perineum, lower abdomen, mons | F | 20–30 | Hispanic | Obesity class II | 3 |
| HS5 | right axilla | F | 20–30 | White | Overweight | 2 |
| HS6 | lower abdomen, mons pubis, bilateral groin | F | 20–30 | White | Obesity class III | not assigned |
| HS7 | bilateral axilla | F | 40–50 | African American | Obesity class II | 3 |
| HS8 | abdomen, bilateral thighs, groin, mons | F | 30–40 | African American | Obesity class II | 3 |
| HS9 | bilateral axilla | F | 40–50 | African American | Obesity class II | 3 |
| HS10 | bilateral axilla | F | 20–30 | African American | Obesity class II | 3 |
| HS11 | bilateral buttocks, sacrum | M | 40–50 | African American | Normal weight | 3 |
| HS12 | lower abdomen, left labia, groin and thigh | F | 40–50 | African American | Obesity class II | 3 |
| CNTRL1 | axilla | F | 40–50 | African American | Obesity Class II | NA |
| CNTRL2 | axilla | F | 20–30 | Asian | not available | NA |
| CNTRL3 | axilla | F | 20–30 | White | Normal Weight | NA |
| CNTRL4 | axilla | M | 60–70 | White | Overweight | NA |
| CNTRL5 | axilla | F | 20–30 | White/Hispanic | Overweight | NA |
| CNTRL6 | axilla | F | 20–30 | Asian | Normal Weight | NA |
| CNTRL7 | axilla | M | 30–40 | Asian | Overweight | NA |
| CNTRL8 | axilla | F | 20–30 | Asian | not available | NA |
| CNTRL9 | axilla | F | 20–30 | Asian | not available | NA |

(Supplementary Fig. 3e), with subcluster 1 being distinctly inflammatory compared to the others, while subcluster 2 appears highly fibrotic (Supplementary Fig. 3f, g). This matches work from van Straalen et al., who found two distinct populations of fibroblasts in HS; one inflammatory and one fibrotic[24]. NCSTN and other GS subunit mRNA expression is not altered. Finally, to test for global transcriptome changes, we applied siRNA to NCSTN to normal epidermal keratinocytes (NHEKs) or normal fibroblasts and performed bulk RNAseq. After NCSTN loss, fibroblasts show greater fold change response than keratinocytes (Fig. 2f). Finally, we compared cultured HS fibroblasts to cultured foreskin fibroblasts treated with NCSTN siRNA. Likely because of the multifactorial nature of HS, as well as the different sites of origin and ages of the fibroblasts, we did not find a complete overlap in gene expression. We instead found a partial overlap with some notable shared transcriptional changes (Supplementary Fig. 3h). Taken together, these results confirm fibroblasts as key signalers of inflammatory signaling in HS and sensitive to NCSTN loss.

### NCSTN loss in fibroblasts potentiates TNFα-induced CXCL8 mRNA/IL8 protein production

Although NCSTN is a known monogenic defect in HS, the mechanism by which it leads to disease is understudied. Given the loss of NCSTN in fibroblasts of sporadic HS and the strong CXCL signaling of fibroblasts in HS, we hypothesized that NCSTN loss primes CXCL signaling in fibroblasts. We therefore investigated how NCSTN loss in normal fibroblasts affects innate immunity. A screen of immunity-related transcripts in the above scRNA datasets identified CXCL8 mRNA

increases in siNCSTN fibroblasts compared to scRNA controls, especially under TNFα stimulation conditions (Supplementary Fig. 4a, b). Given the therapeutic benefit of TNFα inhibition in HS, we focused on TNFα stimulation of fibroblasts in subsequent experiments. CXCL8 mRNA is increased in siNCSTN fibroblasts treated with TNFα compared to scRNA fibroblasts by mRNA (Fig. 3a, knockdown validation in Supplementary Fig. 4c). IL8 protein is elevated after NCSTN siRNA as detected by both ELISA (Fig. 3b) and western blot (Fig. 3c). Consistent with its role in inflammation, IL8 increases with TNFα, but also with siNCSTN in quantification of western blots (Fig. 3d). These results demonstrate the priming of inflammatory IL8 production with NCSTN loss.

To identify one mechanism of how NCSTN loss potentiates IL8 production, we investigated whether the NFkB pathway is primed with NCSTN loss. As IL8 is downstream of TNFR1 and NFkB in this signaling pathway, we searched for altered phosphorylation of the p65 subunit of NFkB. We find that Ser529 phosphorylation of p65 is elevated after NCSTN loss (Fig. 3e, representative western blot, Fig. 3f quantification of three separate blots). On western blot, HS fibroblast lysates demonstrate similar results of increased IL8 and p-NFkB (Ser529) on western blot (Supplementary Fig. 4d). p-NFkB (Ser529) localizes to the nucleus in siNCSTN fibroblasts more strongly than scRNA fibroblasts (Fig. 3g; quantified in 3 h). These results demonstrate NCSTN loss leads to greater phosphorylation and nuclear translocation of p65 in HS fibroblasts.

We next demonstrate that HS fibroblasts mirror this inflammatory potentiation in vitro. HS fibroblasts have higher IL8 signal by IF (Fig. 3i;

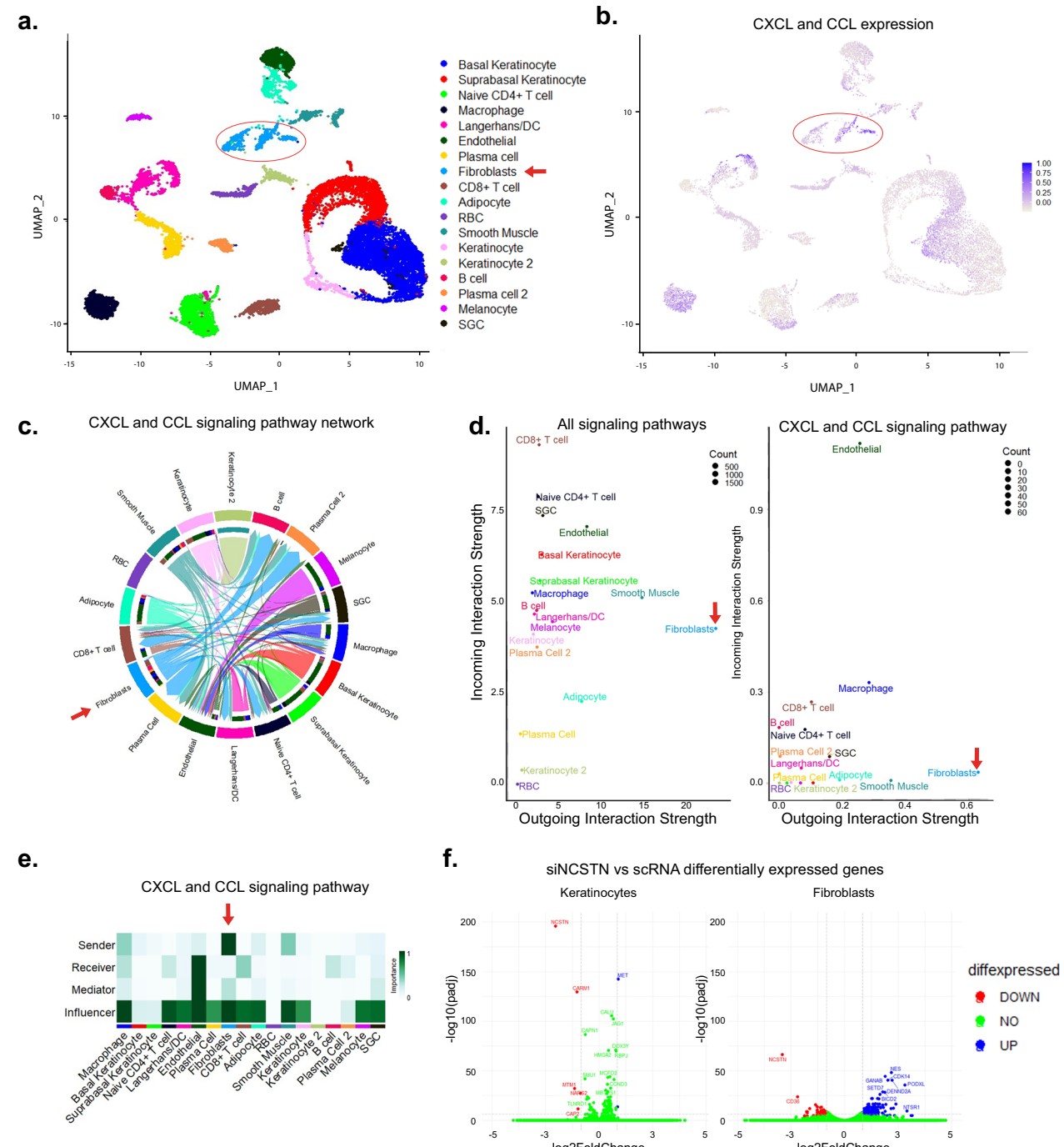

**Fig. 2 | Fibroblasts are the strongest immune signalers in HS by scRNAseq meta-analysis. a** After filtering and normalization of two HS datasets (GSE154775[25] and GSE175990[23]), cells from HS and CNTRL samples cluster into 18 cell identities; fibroblasts are indicated in blue and circled in red. **b** CXCL and CCL signaling is significantly elevated in fibroblasts. **c** Fibroblast originating CXCL and CCL signals are dominant and signal to plasma cells and CD8 + T cells as visualized in a chord diagram. **d** Among all signaling pathways (left) as well as CXCL and CCL specifically (right), fibroblasts are the most prolific senders of signals, while receiving relatively little immune signals. **e** Fibroblasts rank as the strongest senders of CXCL and CCL immune signaling. **f** After siRNA knockdown of NCSTN, fibroblasts demonstrate a larger fold change of affected genes than keratinocytes in bulk RNA seq (*n* = 5). Source data are provided as a Source Data file.

quantified Fig. 3j) when treated with TNFα compared to normal axillary controls (individual donor cells Supplementary Fig. 4e; pooled HS and CNTRL Supplementary Fig. 4f). Additionally, HS fibroblasts also have higher p-NFkB (Ser529) nuclear localization than normal axillary controls by IF (Fig. 3k, quantified Fig. 3l) (individual donor cells Supplementary Fig. 4g; pooled HS and CNTRL Supplementary Fig. 4h). We also correlated this finding in patient tissue samples by IF; HS skin tissue has higher p-NFkB (Ser529) in fibroblasts compared to normal

axillary tissue (Supplementary Fig. 4i, j) and higher IL8 (Supplementary Fig. 4k, l). This overlap between IL8 and p-NFkB (Ser529) is exclusive to fibroblasts in HS skin; despite p-NFkB (Ser529) expression in other cell types. The high IL8 expression in HS dermal fibroblasts and the known role of IL8 to recruit neutrophils suggest that IL8 might correlate with the presence of neutrophils in HS. We confirmed neutrophil accumulation in the dermis of HS skin, even in late-stage disease, typically dominated by lymphocyte activity (Supplementary Fig. 4m). Finally,

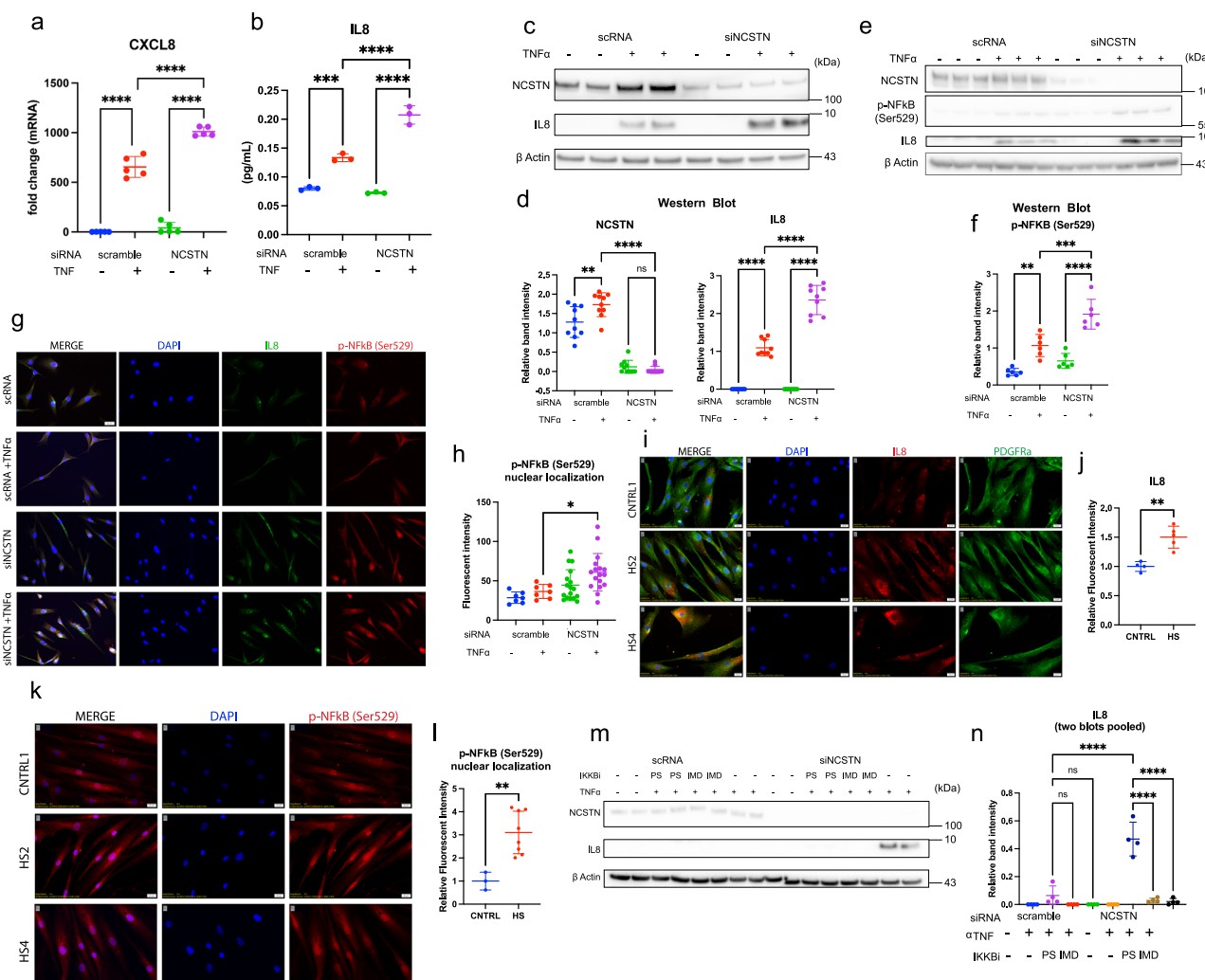

**Fig. 3 | Nicastrin loss potentiates TNFα induced IL8.** TNFα (25 ng/mL for 48 h) treatment to siNCSTN fibroblasts induces greater CXCL8 mRNA and subsequent IL8 protein production as measured by RT-qPCR (**a**) ($p < 0.0001$ $n = 5$), by ELISA (**b**) ($p < 0.0001$, $n = 3$), by western blot (**c**), and quantification (**d**); $p < 0.0001$, $n = 10$ over 4 experiments. **e–h** The potentiated IL8 secretion by NCSTN siRNA correlates to phosphorylation at the Ser529 site of NFκB, as in western blot (**e**) and quantification (**f**) ($p = 0.0002$, $n = 6$ over 2 experiments). The potentiated p-NFκB from NCSTN siRNA localizes to the nucleus as visualized by IF (**g**) and is significant in quantification (**h**); $p = 0.0336$, $n = 7$ over 2 experiments. **i–l** HS fibroblasts recapitulate the findings of Nicastrin siRNA in (**a–h**). TNFα induces more IL8 in HS fibroblasts than CNTRL fibroblasts as detected by IF (**i**) and quantification (**j**); $p = 0.0017$, $n = 3$ CNTRL and 5 HS over 2 experiments. HS fibroblasts express more p-NFκB (Ser529) (**k**) and have higher nuclear localization than CNTRL fibroblasts (**l**) ($p = 0.0046$, $n = 3$ CNTRL and 6 HS over 2 experiments). The priming of IL8 by NCSTN loss is reduced after the application of the Ser529 NFκB kinase IKKB inhibitors PS1145 and IMD0354 (5 µM for 48 h concurrently with TNFα) As detected by Western blot (**m**) and quantification (**n**); PS: $p < 0.0001$, $n = 4$ over two experiments, IMD: $p < 0.0001$, $n = 4$ over two experiments). Quantification values are expressed as mean ± SEM, $p$ values based on one-way ANOVA tests or two-sided Student's $t$ tests. Source data are provided as a Source Data file.

we show phosphorylation of NFkB at the Ser529 site is essential for IL8 production by utilizing two inhibitors of the responsible kinase, IKKB[64,65]. Both small molecule inhibitors of IKKB, PS1145 and IMD0354, ameliorated high IL8 in siNCSTN + TNFα-treated fibroblasts by western blot (Fig. 3m representative western blot; Fig. 3n quantification of two pooled blots). This data shows inflammatory potentiation when NCSTN is lost in fibroblasts, either experimentally manipulated or as part of a disease phenotype, and that it is specifically downstream of NFkB signaling.

**p-EDs are abundant in HS skin and fibroblasts and induce NCSTN loss and IL8 potentiation in healthy fibroblasts**
Associations of EDCs and UPF with HS support the experimental investigation of p-ED effects relevant to HS[18,19]. We hypothesized that p-EDs would be elevated in the skin of HS patients, given the known dietary associations of UPF and HS, and especially with newer survey data showing higher plastic use in an HS patient population[19].

We tested HS skin tissue by MSI for the presence of p-EDs. We first analyzed spectra to generally query data quality. Raw spectra for the parent ions from the overall spectrum for all of the HS samples, as well as the overall average spectrum for all of the CNTRL samples, are shown in Supplementary Fig. 5a. For quality control measures, we also observed ADP to be more intense in CNTRL samples when compared to HS samples. In addition, NADH was more intense in CNTRL samples when compared to HS as well. ATP was detected with similar intensities in both sets of samples (Supplementary Fig. 5b). Arachidonic Acid, a known metabolite critical in inflammatory processes, amongst others, was detected at much higher levels in the HS samples than CNTRL (Supplementary Fig. 5c). Parent ions for metabolite candidates G3P or DHAP, glucose, and pyruvate were all observed, with increased intensity for G3P/DHAP and pyruvate in HS samples when compared to CNTRL samples (Supplementary Fig. 5d). These changes suggest success of MSI in general quality and the ability to differentiate HS from normal samples.

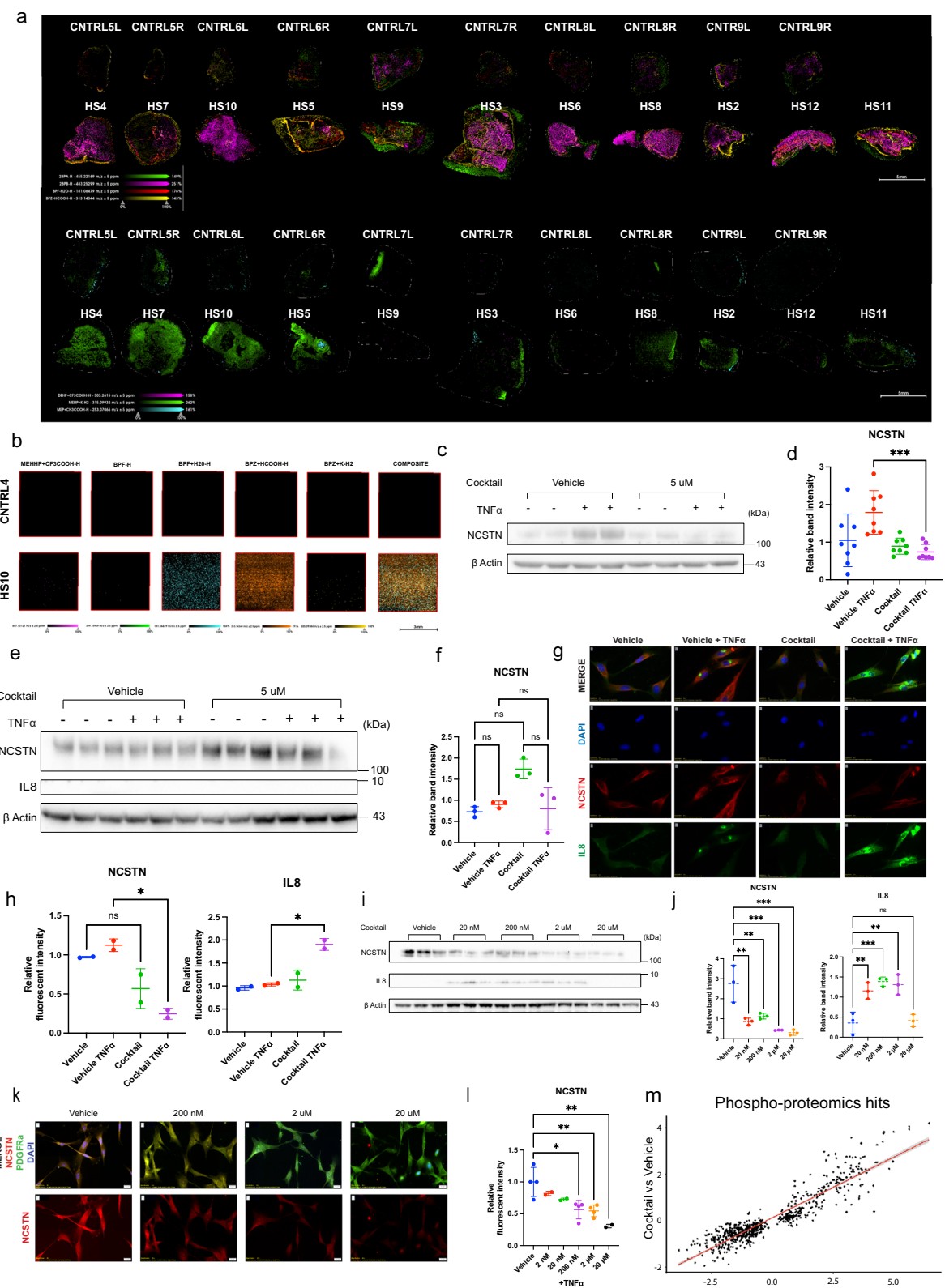

Concentrations of p-EDs adducts appeared elevated in HS compared to normal axillary control skin (Fig. 4a; bottom 2 rows). The parental phthalate DEHP was elevated in HS, as were its metabolic breakdown products, including MEHP and MEP, suggesting systemic exposure. Phthalates are known to directly bind to glycogen-debranching enzyme to perturb glucose metabolism, and while our protocol was not optimized for glycolytic metabolites, several

glycolytic metabolite candidates were observed. However, several of these species have the same molecular formula, so further LC-MS-based experiments would need to be conducted to confirm their identities. Perhaps more consistently elevated in HS skin than the phthaltes were elevations of the adducts of bisphenol subtypes BPA, BPB, BPF and BPZ (Fig. 4a; top 2 rows). Statistical analysis of MSI between patient and control groups is available in Supplementary

**Fig. 4 | p-ED compounds are abundant in HS skin and fibroblasts and induce NCSTN loss and IL8 production in normal fibroblasts.** HS skin contains high intensity of p-EDs compared to normal axillary controls as visualized by mass spectrometry imaging (MSI) at the tissue level (**a**; top bisphenols, bottom phthalates), and in ex vivo cultured cells (**b**). A 200 nM mixture of 8 p-EDs (7 days exposure; 48 h of 25 ng/mL TNFα) inhibits NCSTN expression as visualized by western blot (**c**) and quantification (**d**) (*p* = 0.0004, *n* = 8 over 3 experiments) in CNTRL fibroblasts, but not in keratinocytes (**e**, **f**); *p*=ns, *n* = 3 per condition). p-ED cocktail (5 μM for 7 days exposure; 48 h of 25 ng/mL TNFα) inhibits NCSTN and primes TNFα induced IL8 expression as visualized by IF (**g**) and quantification (**h**); p <.0001, *n* = 2 over 2 experiments). **i–l**, p-ED mixture (7 days; 48 h of 25 ng/mL TNFα)

induces a dose responsive loss of NCSTN as found in western blot (**i**), and quantification (**j**); *p* = 0.0003 overall, *n* = 3 per dosage). IL8 increase compared to vehicle control is also found by western blot for concentrations 20 nM–2 μM (**j**); *p* = 0.0002 overall, *n* = 3 per dose). IF (**k**), and IF quantification (**l**) recapitulate the dose-responsive relationship of NCSTN loss to p-ED mixture starting at 200 nm (*p* < 0.0001 overall, *n* = 4 for vehicle, 200 nM, and 2 μM, and *n* = 2 for 2 nM, 20 nM, and 20 μM over 2 experiments). **m** Proteomic analysis shows phosphorylated protein hits for HS vs CNTRL are recapitulated by p-ED cocktail treatment vs Vehicle (*R²* = 0.8495, *p* < 2.2 ×10⁻¹⁶). Source data are provided as a Source Data file. Quantification values are expressed as mean ± SEM, *p* values based on one-way ANOVA tests or two-sided Student's *t* tests.

Fig. 6a, b, and H&E stains of the tissue specimens used are available in Supplementary Fig. 6c. Given the cellular memory of NCSTN loss of HS fibroblasts, we next tested by MSI cultured fibroblasts from HS or control tissue. High p-ED adducts in HS samples persist in vitro; fibroblasts extracted and passaged twice in vitro from one normal axillary sample and one HS sample show the same effect as in tissue, where HS fibroblasts have higher p-ED adduct concentration than normal axillary fibroblasts (Fig. 4b). In the case of cultured HS fibroblasts, phthalates were not detected to a high degree, but instead strong signals for multiple bisphenols, most notably BPZ and BPF adducts, were measured. This could indicate either a predominance of bisphenols in HS skin over phthalates or represent a more persistent presence of bisphenols. Alternatively, it could be a combination of both persistence and abundance. Quantification of ex vivo MSI is shown in Supplementary Fig. 6d. This data shows not only do HS patients have higher intensity of p-ED adducts in their skin, but the p-ED adducts are physically located within dermal fibroblasts and persist over at least three cell divisions.

We next aimed to test the functional importance of p-EDs rather than simply their correlative detection in HS. To examine whether p-EDs have an effect, we treated normal fibroblasts with a mixture of 8 detected p-EDs (5 bisphenols and 3 phthalates) for a total concentration of 5 μM with and without TNFα. In the presence of TNFα in all samples, p-ED treatment inhibited NCSTN protein exression compared to vehicle in normal fibroblasts (Fig. 4c, quantified Fig. 4d), similar to that found in HS physiology. Interestingly, this effect is not seen in keratinocytes (Fig. 4e, quantified Fig. 4f). Keratinocytes are thus resistant to p-ED effect on NCSTN protein; this mirrors the findings in HS patient tissue stains with low NCSTN in dermal fibroblasts but not in keratinocytes (Fig. 1a). The loss of NCSTN in fibroblasts treated with p-ED mixture and TNFα occurs at both the protein level and the mRNA level (Supplementary Fig. 7a). Confirming a likely systemic biologic connection, p-EDs inhibited the transcription of all GS subunit genes and isoforms besides PSEN1 (Supplementary Fig. 7a). p-EDs +TNFα increased CXCL8 and TNFα mRNA compared to vehicle + TNFα control (Supplementary Fig. 7b, c). In the context of how NCSTN loss primes IL-8, these results highlight the relationship between p-EDs, NCSTN protein loss, and inflammatory potentiation.

In vitro IF imaging of normal fibroblasts treated with 2 μM p-ED mixture and TNFα confirmed these findings to show low NCSTN protein and elevated IL8 production (Fig. 4g, quantified Fig. 4h). Extended data is provided in Supplementary Fig. 7d, e. We also established a dose responsive relationship between p-ED mixture concentration and NCSTN protein (all with TNFα treatment); by western blot, NCSTN protein loss was dose responsive from 20 nM to 20 μM treatment of p-EDs, with IL8 response being higher consistently from 20 nM to 2 μM and dropping off at 20 μM (Fig. 4i, quantified Fig. 4j). IL8 is mostly uniformly elevated even low levels of NCSTN loss, suggesting elevated IL8 in the setting of NCSTN loss is not dose responsive.

Given 20 nM of p-ED mixture still showed NCSTN loss, we sought to determine the lowest concentration at which we could observe a loss of NCSTN with the p-ED mixture and verify these findings in a different experimental protocol. In universally applied TNFα, we dosed

normal fibroblasts from 2 nM to 20 μM and performed IF (representative images Fig. 4k), in our experience more sensitive than western blot. When multiple repeats are pooled, statistical loss of NCSTN starts at 200 nM (Fig. 4l). However, within individual experiments, dosing as low as 2 nM p-ED cocktail (equivalent to 250 pM of each compound) caused a significant loss of NCSTN protein (Supplementary Fig. 7f). Given the somewhat greater inhibition of NCSTN by p-EDs at the protein level compared to the mRNA level, we probed for potential post-transcriptional effects of p-EDs on NCSTN. To determine if proteosome function or other protein synthesis contributes to NCSTN protein loss in the context of p-ED cocktail and TNFα treatment, we used 100 nM of the proteosome inhibitor, MG-132, and 0.25 μg/mL puromycin as a ribosome inhibitor. MG-132 did not rescue NCSTN protein loss, suggesting NCSTN is not directly degraded by the proteosome in this scenario. Interestingly, puromycin did significantly rescue NCSTN levels back to vehicle control levels (Supplementary Fig. 7g, h). This suggests the translation of another protein may be responsible for the degradation of NCSTN. Alternatively, different concentrations and time periods of proteosome or ribosome inhibitors might be required to assess an effect.

We treated normal fibroblasts with either our p-ED mixture or a gamma secretase inhibitor (DAPT) to determine how similar the overall transcriptional profile was between the two treatments that modeled HS biology. Remarkably, bulk RNAseq analysis demonstates an almost exact correlation between DAPT, a gamma secretase inihibitor (GSI) and p-ED mixture treatment (Supplementary Fig. 7i), with *R²* = 0.8968. This suggests p-ED mixture exposure on fibroblasts is biologically and functionally congruent with direct gamma secretase inhibition, further underscoring the importance of p-EDs in HS.

Finally, we performed proteomics on HS, CNTRL, control foreskin fibroblasts, and p-ED treated control fibroblasts to determine if p-ED mixture treatment could recapitulate dynamic protein changes seen in HS vs CNTRL proteomics. As shown in Fig. 4m, phosphorylated protein hits significantly changed in the HS vs CNTRL comparison were closely matched by the p-ED treatment vs control group. The *R²* = 0.8495 (*p* < 2.2 × 10⁻¹⁶), indicating a strong relationship between the two comparisons, further emphasizing the similarities between our HS models of primary HS cells and p-ED treated control cells.

This data shows how environmentally relevant concentrations of p-EDs (down to 250 pM per compound) causes loss of NCSTN protein. Consistent with this, we demonstrate that p-EDs can recapitulate the phospho-proteome found in HS fibroblasts. Altogether, these results show high biologic concordance between p-ED exposure, NCSTN siRNA treatment, and HS phenotype of dermal fibroblasts.

## Discussion

Given the speed of industrialization, the rapidly evolving landscape of chemical exposure to humans through food is certain to have health impacts. Confirming a parallel increase in the incidence of sporadic HS and UPFs, we find an association between p-EDs and disease. We demonstrate that sporadic HS recapitulates the pathogenesis of monogenic HS through shared decrease in NCSTN protein production and primed innate immunity. We also show that p-EDs are elevated in

the skin of patients with HS and inhibit NCSTN protein expression at nanomolar concentrations to promote inflammation. These results suggest that reducing exposure to p-EDs may decrease HS severity, and suggest the importance of further investigation on mechanisms, including the relevance of these findings to other diseases associated with GS dysfunction, such as Alzheimer's disease.

In this study, we demonstrate through hypothesis-driven approaches a novel contributory mechanism to HS pathogenesis, likely working in parallel with other disease-promoting events in this multifactorial disease. NCSTN protein expression is lost in dermal fibroblasts, and this loss is maintained ex vivo in extracted cells. Meta-analysis of scRNAseq studies of HS supports fibroblasts as dominant immune signalers in disease pathogenesis, with particularly elevated CXCL cytokines responsible for immune cell recruitment. Modeling HS fibroblasts using an siRNA knockdown of NCSTN in normal fibroblasts reveals a biology hypersensitive to TNFα, with elevated phosphorylation of NFkB and IL-8 protein expression. Finally, we link dietary factors, a known risk factor for HS, with these molecular discoveries. Several p-EDs, known to be present in UPFs correlated with HS incidence, are elevated in HS skin compared to site-matched controls. Ex vivo cultured HS dermal fibroblasts even sequester p-EDs, but matched healthy axillary fibroblast controls do not. Exposing normal cells in vitro with a mixture of these p-EDs at environmentally relevant nanomolar levels demonstrates remarkably similar biological responses to HS patient cells. Fibroblast-specific NCSTN protein loss and primed IL8 production are shared features of p-ED-treated and HS fibroblasts, with NCSTN siRNA recapitulating these findings. This important overlap is highly indicative of a likely contributory role of p-EDs and NCSTN loss in sporadic HS pathogenesis. However, our work also raises questions as to what co-exposures or genetic susceptibilities might make p-EDs more potent in HS pathogenesis in select individuals.

Given the centrality of tunnel formation to the clinical features of HS, it is important to question if the present work bears relevance. Tunnel formation is actuated by dysfunctional keratinocytes. While we do not directly connect p-EDs and dysfunctional keratinocytes to tunnel formation in this study, fibroblasts are major contributors to and drivers of keratinocyte differentiation, function, and inflammation. As others[24], we noted several findings in published scRNAseq datasets that may provide additional context. We identified versican positive—likely dermal papillae or hair follicle associated—fibroblasts (VCAN+, MMP3+, MMP1+) to be the main sources of both CXCL13 and MMP3 in HS patients. This subcluster represents the inflammatory fibroblast population and is also responsible for producing the majority of IL8, after macrophages, in HS patient tissue samples (subcluster 1, Supplementary Fig. 3e–g). In this regard, our findings on fibroblasts could potentially direct keratinocytes towards tunnel formation. For example, p-ED exposed fibroblasts have higher IL8 production, which is also reflected in our in vivo findings. Keratinocytes express the IL8R and have been shown to be chemoattracted to IL8[66] and exogenous IL8 application induces migration of HaCaT cells[67]. Taken together, p-ED exposed NCSTN deficient fibroblasts produce more IL8 under inflammatory conditions, which may stimulate increased keratinocyte migration, contributing to sinus tunnel formation in disease pathogenesis. Future work should test whether hair follicle keratinocytes might be more sensitive to IL8 to initiate tunnel formation, for example.

An important question is why HS lesions occur primarily in intertriginous areas, their connection to hair follicle occlusion, and how these features and p-EDs might be connected to an autoinflammatory condition. We favor several overlapping hypotheses. Given the high sweat output in the axillae, sweat exposure itself might be a driver of early HS pathogenesis. p-EDs are lipid-soluble molecules, and eccrine and apocrine glands could be secreting p-EDs in sweat, which would be reabsorbed in intertriginous areas, enhanced by occlusion of opposing skin. This might induce an additional avenue of concentrated p-ED exposure. This may also explain the involvement of hair follicle fibroblasts; a common area of accumulation for topical chemicals is the hair shaft canal. After sweating, p-EDs might concentrate in hair shaft canals and more specifically activate hair follicle associated cell populations, perpetuating a cycle of inflammatory signaling. Additionally, a completely separate mechanism may be induced in the hair follicles at this time with follicular occlusion promoting inflammation and HS pathogenesis in parallel to p-ED biology. We and others have explored this question, but future studies should include further mechanistic experiments to address these hypotheses[18,19].

Human exposures to microplastics and p-EDs have dramatically increased over the years. Several EDCs are proposed as risk factors for diseases such as diabetes, infertility, metabolic disorders and cancer[68–70]. For example, microplastic presence in atheromatous plaques increases risk of adverse events, including myocardial infarction and stroke, in cardiovascular disease patients[71]. Even more recently, microplastics were identified at higher concentrations in human brains with dementia than control, a finding bearing particular relevance to our own work since GS dysfunction is implicated in Alzheimer's dementia[72]. Previously, many HS patients are known to consume UPF-heavy diets, likely exposing them to high levels of p-EDs, but the clinical relevance was unknown. We have identified the molecular mechanism of p-EDs causing HS-like phenotypes in vitro. Our data suggest p-EDs, in addition to microplastics, should be investigated further for disease associations, especially diseases with an incidence increase related to processed food and plastic production.

This study raises many important questions for future study. For example, we used a mixture of eight p-EDs to best represent exposure in the real world in our study. However, it is possible that some of these compounds may have differential effects when used in isolation and should be studied independently. Furthermore, many p-EDs are nearly ubiquitously present in human diets, blood, and serum. However, we found very low levels of p-EDs in normal human skin compared to HS skin, raising the critical question of why and how some people (such as HS patients) retain or sequester p-EDs in local tissue while others (normal controls) do not. Although differences in p-ED total consumption is likely an important explanation, p-ED body wide trafficking, cellular absorption and cellular elimination are highly important area of future study. For example, genetic differences might modify any combination of these trafficking events to enhance susceptibility to p-EDs. Similarly, gut permeability differences might exist in HS versus controls, consistent with inflammatory bowel disease as a known comorbidity. We made an effort to ensure we had controls with overweight or above BMIs to better match our experimental population. However, it was not feasible to match dietary habits given the large variety and lack of an easy, practical, and well-established quantitative index of nutritional habits. It is a limitation of this study, and further work should explore in more detail the exact and complete effects of different nutritional composition on p-ED exposure and HS pathogenesis. We were also unable to perfectly match ethnicities to our patient population, as only one control was African American. Finally, future work should overexpress NCSTN in fibroblasts, to measure the degree of rescue of the HS or p-ED exposure phenotype.

Our focus on fibroblasts, GS, and p-EDs, and TNFα does not preclude the likely importance of parallel pathways that promote HS disease. We used TNFα treatment as an immune stimulator and IL8 as a marker of inflammatory potentiation in this manuscript. Most of the patients in our study had at least a partial, if temporary, response to TNFα inhibition (Humira) in the early stages of their treatment and disease. But many of our patients did eventually fail Humira, suggesting study of other potential inflammatory pathways, like IL-17, is essential to better understand global HS pathogenesis. With the recent success of Cosentyx and Bizekizumab, both IL-17 inhibitors, in treating

HS, further studies should also explore how IL-17 potentiates HS pathogenesis. It is possible IL-17 acts completely independent of p-ED exposure, or works on a different cell type than our study, such as keratinocytes. Additionally, we cannot rule out the possibility of GS-independent functions of NCSTN may be involved in our inflammatory cascade hypothesis outlined here. Similarly, GS biologists have long known loss of one GS subunit often causes destabilization and subsequent degradation of the other components. As such, we cannot exclude the possibility that the NCSTN loss with p-EDs is partially due to the loss of other GS subunit proteins. We were also not able to reliably detect the immature, non-glycosylated version of NCSTN in all our experiments, and analyzed total NCSTN protein in our results. However, glycosylation of NCSTN to its mature, active form is a critical component of GS biology, and this and other post-transcriptional mechanisms should be studied in detail in the context of p-ED exposure in future work. The work of all investigators in the HS field is essential to fully understand this highly multifactorial disease.

Finally, it will be important to investigate the implications of the present work for other diseases besides HS. For example, GS signaling disturbance is likely a critical consequence of NCSTN loss, given that GS-independent functions for NCSTN are not currently appreciated. GS dysfunction is well studied in the pathophysiology of familial Alzheimer's disease. Since we have shown a loss in the NCSTN subunit of GS with p-ED exposure, and p-EDs and microplastics have been found in the brain, p-EDs might alter inflammatory signaling in the brain and might modify Alzheimer's risk.

In summary, this work highlights the importance of immediately counseling HS patients to minimize p-ED exposure as a potential treatment for their disease. While we show strong evidence of possible causation of p-EDs inducing HS-like phenotypes in our system, the only definitive proof of causation would be the complete removal of p-EDs from HS patients as a cure for the disease, which is not feasible. Further work in the field is required for continued study of p-EDs or other exposures that might promote HS and how we may mitigate it. Regardless, this work stresses the underrecognized importance of environmental exposures in the pathogenesis of HS.

# Methods

## Cell culture
Fibroblasts were seeded in plates or dishes with FGM™-2 Fibroblast Growth Medium-2 BulletKit™ (Lonza CC-3132). Fibroblasts used in siRNA experiments and cocktail treatments were fetal fibroblasts extracted directly from foreskin. Cells were fed with fresh media every 3 days and split for maintenance or experimental seeding using prewarmed Trypsin (Lonza CC-5012) for 5 min and neutralized with an equivalent volume of TNS (Lonza CC-5002). Cell suspensions were spun down at 770 RCF at 4 °C to pellet cells, and supernatant removed. Cells were resuspended in fresh FGM, counted, and seeded. Keratinocytes were cultured identically, but with KGM™ Gold Keratinocyte Growth Medium BulletKit™ (Lonza 00192060) media and trypsinization was performed for 7 min. For TNFα incubations, 25 ng/mL of TNFα was added to the media for 48 h.

Cocktail was composed of BPA, BPB, BPS, BPZ, DEHP, MEHP, and MEP in equivalent concentrations with DMSO vehicle. Vehicle DMSO control treatments were equivalent to the percentage of DMSO in cocktail treatments. For dose response curves, DMSO vehicle percentage was equivalent in all different concentrations and in Vehicle only control. It is important to minimize the exposure of plastics to cells and reagents during the experiment, such as avoiding maintaining DMSO in plastic eppendorfs or excessive UV light disinfectant exposure to tissue culture contacting plastics.

For cocktail incubations, cells were seeded in normal media on day 0 and allowed to attach overnight before being switched to cocktail-containing media the next day. Cocktail mixtures were premixed to include all compounds at equivalent concentrations, then added to media to reach their final concentration. TNFα was added to the media for the last 48 h of the experiment, without changing the media.

Bisphenol A (BPA) MilliporeSigma 1075892
Bisphenol B (BPB) MilliporeSigma 50877
Bisphenol F (BPF) MilliporeSigma 51453
Bisphenol S (BPS) MilliporeSigma 43034
Bisphenol Z (BPZ) MilliporeSigma 450421
Monoethyl Phthalate (MEP) MilliporeSigma SMB00941
Bis-(2-ethylhexyl) Phthalate (DEHP) MilliporeSigma 36735
Phthalic Acid mono-2-ethylhexyl Ester (MEHP) MilliporeSigma 796832

## siRNA Knockdowns
Either Santa Cruz Nicastrin siRNA (sc-36063) or ThermoFisher Nicsatrin siRNA (s23707) and either control siRNA-A (sc-37007) or ThermoFisher *Silencer*™ Select Negative Control No. 1 siRNA (4390843) were used for knockdown experiments. A final concentration of 50 pmol of siRNA was used. Lipofectamine™ RNAiMAX Transfection Reagent (ThermoFisher 13778075) was used with Gibco™ Opti-MEM™ I Reduced Serum Medium (ThermoFisher 31985062) and transfection performed as in the RNAiMAX Reagent protocol. Briefly, total amount of RNAiMAX was diluted in Opti-MEM™ and siRNA also diluted in Opti-MEM™; the two solutions were mixed and incubated at room temperature for 5 min to allow complexes to form. Complexes were then added to plates. Media was changed the morning after transfection.

## Immunofluorescence
*Antibodies – Primary:*
Nicastrin (h): Santa Cruz sc-376513 1:500 (Alexa Fluor® 594 conjugated or unconjugated)
PDGFRα (h): Abcam ab203491 1:500 (Alexa Fluor® 488 conjugated or unconjugated)
IL8 (h): 1:300 Thermo Fisher M801
p-NFκB (h): 1:250 ThermoFisher 44-711G
*Antibodies – Secondary:*
Alexa Fluor® 488 Goat Anti-Rabbit
Alexa Fluor® 488 Goat Anti-Mouse
Alexa Fluor® 594 Goat Anti-Rabbit
Alexa Fluor® 647 Goat Anti-Mouse
Alexa Fluor® 647 Goat Anti-Rabbit

For tissue slides, paraffin-embedded sections were rehydrated using the following series for 10 min each: Xylenes (x2), 100% ethanol (EtOH) (x2), 90% EtOH, 80% EtOH, 50% EtOH, dH₂O. After, slides were submerged in 1X Antigen Retrieval solution (BioRad BUF025B) and heated in a pressure cooker at 120 °C for 7 min. Slides were cooled slowly with a slow dH₂0 drip. Slides were laid flat, and a barrier drawn around the tissue samples with a hydrophobic barrier pen (Thermo-Fisher R3777). If permeabilization was required (stains for IL8, p-NFκB), 100% ice-cold MeOH was used for 10 min followed by 2 PBS washes for 5 min each. Otherwise, slides were incubated with blocking solution (10% Normal Goat Serum, 5% BSA in PBS) for 30 min at room temperature, then incubated overnight with primary antibody diluted in blocking solution at 4 °C in the dark. The following morning, slides were washed 3 times with PBS for 5 min each, then incubated with secondary antibody if necessary for 1 h at room temperature in the dark diluted in blocking solution. Slides were then washed 3 times with PBS for 5 min each. Finally, slides were mounted with ProLong™ Gold Antifade Mountant (ThermoFisher P36931) and coverslips placed and sealed with clear nail polish. Nail polish was allowed to fully dry before slides were imaged in VS200 Olympus Research Slide Scanner.

For in vitro immunofluorescence, cells were cultured in chamber cell culture slides (CellTreat 229162). At time for analysis, media was aspirated and chambers washed with PBS for 5 min. Cells were fixed with 4% Paraformaldehyde in PBS for 15 min at room temperature then

washed 3 times for 5 min each with PBS. If permeabilization was required (stains for IL8, p-NFκB), 100% ice cold MeOH was used for 10 min followed by 2 PBS washes for 5 min each. Otherwise, slides were incubated with blocking solution (10% Normal Goat Serum, 5% BSA in PBS) for 30 min at room temperature, then incubated overnight with primary antibody diluted in blocking solution at 4 °C in the dark. The following morning, slides were washed 3 times with PBS for 5 min each, then incubated with secondary antibody if necessary for 1 h at room temperature in the dark diluted in blocking solution. Slides were then washed again 3 times with PBS for 5 min each. Finally, chamber slide dividers were removed, and slides were mounted with ProLong™ Gold Antifade Mountant (ThermoFisher P36931) and coverslips placed and sealed with clear nail polish. Nail polish was allowed to fully dry before slides were imaged in VS200 Olympus Research Slide Scanner. We generally find that immunofluoresence was more reliable than western blotting for NCSTN quantification.

### ImageJ immunofluorescence analysis

Three channel TIFs from IF scans were imported into ImageJ (https://imagej.net/ij/) and split into their color channels. For NCSTN fluorescent intensity, the PDGFRα channel threshold was adjusted to identify single fibroblasts. This threshold image was used to analyze particles, which were then projected onto the NCSTN fluorescent channel, and each object (cell)'s average (by area) fluorescent intensity of NCSTN was measured. Each number thus corresponded to the average NCSTN fluorescent intensity of a single cell normalized to its relative size. This data was imported into GraphPad, where an outlier analysis was performed to exclude outliers (often obviously due to extracellular antibody aggregates). The cleaned data were then used to perform ANOVA analysis between samples. Nuclear localization quantification was performed similarly, but DAPI channels were used to define objects (nuclei), which were then overlaid on the p-NFκB channel and fluorescent intensity quantified per nuclei. IL8 quantifications were performed using PDGFRa as the cell identification channel as for NCSTN.

To combine analyses of multiple separate experiments for the same protein (i.e., NCSTN in vitro analysis), fluorescent intensity per cell was further normalized by dividing by the average fluorescent intensity of the control cells/conditions within that experiment, yielding fluorescent intensities per cell centered around 1. These could then be combined across experiments.

When called for, the average fluorescent intensity of all cells in a sample was averaged to create a single data point per sample (i.e., tissue immunofluorescence of HS patients). These averages were then pooled into control and HS and compared using a student's *t* test. Raw data files of fluorescent intensity per cell and original images are available upon request.

### Western blotting
*Antibodies – Primary:*

Nicastrin (h): ThermoFisher MA5-29438 1:1000
Also tested Nicastrin (h): ThermoFisher PA5-17735 1:1000
p-NFκB (h): ThermoFisher PA5-37722 1:500
NFκB (h): ThermoFisher PA5-27617 1:1000
IL8 (h): ThermoFisher PA5-79113 1:500
HES1 (h): ThermoFisher PA5-28802 1:1000
B-Actin (h): ThermoFisher PA1-183-HRP 1:2000
*Antibodies−Secondary:*
Anti-rabbit IgG, HRP-linked: Cell Signaling Technology 7074
Anti-mouse IgG, HRP-linked: Cell Signaling Technology 7076

Cells were cultured for experimental duration on 6-well plates or 10 cm dishes. When ready to harvest protein, cells were washed with cold PBS, then trypsinized for 5 min using prewarmed Trypsin (Lonza CC-5012). Trypsin was neutralized with an equal volume of TNS (Lonza CC-5002), and resulting cell suspension centrifuged at 770 RCF at 4 °C

for 5 min to pellet cells. Cell pellet was washed with ice-cold PBS and centrifuged at 770 RCF at 4 °C for 5 min. Supernatant was aspirated, and final pellet resuspended in RIPA buffer (Millipore Sigma R0278) with protease/phosphatase inhibitor cocktail (Cell Signaling Technology 5872S). Lysates were then spun at 20,000 RCF for 20 min at 4 °C and resulting cleared supernatants transferred to new tube. Remaining debris pellet was discarded. Pierce™ BCA Protein Assay (ThermoFisher 23225) was used to quantify protein lysates, and 20 μg of protein was run on a Bis-Tris gel, which was transferred to a PVDF membrane using a wet transfer tank for 1h. Membrane was stained with Ponceau to check loading, then blocked in 5% NFDM in TBST for 30 min and incubated overnight with primary antibodies at 4 °C. The membrane was washed with TBST 3 times for 5 min each, then incubated with secondary in blocking solution for 1 h at room temperature. Membrane was washed three times for 5 min each with TBST then develop-ED (ThermoFisher 34579) and imaged on a ChemiDoc system. Original unprocessed and uncropped scans are available in Source Data file.

### RT-qPCR
*Taqman Probes:*

GAPDH: Hs02786624_g1
NCSTN: Hs00299716_m1 CXCL8: Hs00174103_m1
HES1: Hs00172878_m1
TNF: Hs00174128_m1
PSEN1: Hs00997789_m1

Cells were grown for experimental duration in 12 well or 6 well plates. At experimental end, cells were washed twice with PBS then lysis buffer from Qiagen RNeasy kit (Qiagen 74136) added to wells. Lysis buffer was pipetted up and down in the well to ensure full lysis of entire well. Lysis solution was transferred to an Eppendorf tube and protocol from RNeasy kit was followed through to yield RNA in water at end of protocol. RNA was quantified using a Nanodrop, and a reverse transcription reaction prepared with 1000 ng of RNA per sample. Applied Biosystems™ High-Capacity cDNA Reverse Transcription Kit (ThermoFisher 4368814) was used for reaction reagents and Thermocycler was set to specifications indicated on the kit manual. Resultant cDNA solution was diluted with an additional 150 μL of RNAse-free $dH_2O$ and mixed thoroughly. qPCR reaction was prepared using TaqMan probes and Applied Biosystems™ TaqMan™ Fast Advanced Master Mix for qPCR (ThermoFisher 4444556) according to product manual. Samples were run in technical duplicates or triplicates on qPCR plates. qPCR reactions were performed on Thermo ABI QuantStudio 5 Real-Time PCR System and analyzed on QuantStudio™ Design and Analysis Software. Raw data was exported to Microsoft Excel for delta CT analysis and final data transferred to Prism GraphPad for statistical analysis and visualization.

### ELISA

Media from fibroblasts treated with TNFα for 48 h was collected and centrifuged for 10 min at 2000 RCF at 4 °C to pellet any cell debris. Supernatant was diluted 1:10000 in Sample Dilutant from IL8 ELISA kit (Abcam ab214030). Kit manual was followed to prepare plate for analysis and plate read on a microplate reader as indicated in the manual. Samples were run in technical triplicate.

### scRNAseq analysis

scRNAseq datasets from two studies (GSE1547755 and GSE1759906), comprising samples from a total of 12 HS patients and 1 healthy control, were processed using the R package Seurat version 4.3.0.

Pre-analysis quality control and cell selection followed standard Seurat guidelines. Briefly, samples were filtered to include cells with RNA counts between 200 and 2500, and mitochondrial transcripts of less than 5% of total RNA transcripts. The RNA expression data of each cell were then normalized and transformed into a logarithmic scale. Studies were integrated with highly variable features were identified to

improve processing efficiency of downstream analysis, and all data scaled so the variance across all cells was 1.

To analyze the quality-controlled data, an unsupervised PCA analysis was performed to cluster cells. Cell cluster identities were determined by calling the Seurat function FindMarkers and manually determining the cell types using the top 5 genes expressed for each cluster.

Signaling analysis was performed using the R package CellChat version 1.6.19. Standard preprocessing was performed, followed by individual signaling analysis of the CXCL and CCL pathways.

To examine fibroblast subpopulations in detail, the fibroblast cluster identified from all samples was extracted into a new object, and the clustering analysis was repeated. To visualize all CXCL/CLL expression in feature plots, a modular score was calculated by summing expression data for all genes beginning with CXCL-, CCL-, and assigning this value as a separate value using Seurat's FindModularScore feature.

## Patient tissue collection and preparation

CNTRL tissue samples (CNTRL 1–9) were obtained as punch biopsies from the axilla of patients in dermatology clinic without HS or other skin conditions in their axilla. Participants were presented with an informed consent document they read. Participants were given an opportunity to ask any questions and then signed the informed consent document before full participation in the study. Participants consented to the study and procedure and were numbed with local anesthetic before undergoing punch biopsy. Patients were compensated $60 for their participation. Tissue samples were collected in CO2 independent media (Gibco 18045088) to maintain tissue viability and immediately transferred to the lab and washed thoroughly with PBS.

HS skin tissue samples (HS 1–12) were obtained from wide excisional surgeries conducted as definitive HS treatment for patients. Participants were presented with an informed consent document they read. Participants were given an opportunity to ask any questions and then signed the informed consent document before full participation in the study. Tissue samples were obtained under sterile conditions and collected in CO2 independent media (Gibco 18045088) to maintain tissue viability. Following collection, samples were thoroughly washed with PBS. Adipose and excess scar tissue were excised and discarded. Subsequently, 8–12 4 mm punch biopsies were extracted from the remaining tissue and proceeded to processing.

For both CNTRL and HS skin samples, ~1/5 of tissue was preserved for Mass Spectrometry (see below section "MALDI and DESI"), 1/5 for histological immunofluorescence (see above section "Immunofluorescence"), and 3/5 for cell extraction for in vitro experiments (see below for further processing).

Collection of CNTRL normal axillary samples was under NA_00033375 "Skin biopsy study to determine gene expression signature of skin" and HS excisional samples under NA_00031269 "Use of Discarded Human Skin Surgery Tissue for Dermatology Research".

## Biopsy sample processing

Fresh human skin biopsy samples were immediately processed under sterile conditions to preserve cellular viability and minimize contamination. Samples were washed with 70% EtOH for 1 min, rinsed with PBS, and treated with 10% antibiotic solution (Penicillin-Streptomycin 10%) for 10 min. To ensure adequate penetration of the antibiotic, samples were gently agitated during incubation. Samples were incubated overnight in Enzyme G in RPMI 1640 Medium (Miltenyi GentleMACS Epidermis Dissociation Kit 130-103-464, Gibco 11875-093) with constant mixing at 4 degrees Celsius. On day 2, the epidermal and dermal layers were separated with sterile forceps under a hood to ensure clean separation of tissue layers. Each layer was processed independently to isolate viable keratinocytes and fibroblasts (see below).

## Epidermal layer processing

Epidermal tissue was collected in warmed trypsin and incubated at 37 degrees, with intermittent vortexing every 3 min for a total duration of 15 min. This step promoted dissociation of epidermal cells while minimizing mechanical shear stress. The trypsin was neutralized with TNS, and the solution was passed through a 70 µM filter to remove any undigested tissue fragments and multicell clumps. The filtrate was centrifuged at $700 \times g$ at 4 °C for 5 min, after which the supernatant was discarded, and the remaining pellet was resuspended in Keratinocyte Growth Medium (KGM). KGM was used to support proliferation and maintain epithelial phenotype. The resuspended cells were immediately plated onto a 10 cm dish, with a media change within 24 h to promote cell viability and proliferation. Cultures were maintained at 37 °C with 5% $CO_2$ and monitored daily for adherence and morphology.

## Dermal layer processing

Dermis was treated with 0.1% collagenase in PBS (Sigma-Aldrich C9891, 9001-12-1) and incubated at 37 °C for 15 min, with vortexing every 3 min to facilitate tissue dissociation. Collagenase digestion helped break down the extracellular matrix, enabling release of fibroblasts. The collagenase activity was neutralized using Dulbecco's Modified Eagle Medium (DMEM, Gibco 11965-092) with 10% FBS, and the resultant mixture was passed through a 70 µm filter into a GentleMACS Dissociator purple cap C-Tube (Miltenyi Biotec 130-093-237). Dermis dissociation was further facilitated utilizing the "skin dissociation" protocol on the GentleMACS dissociator. Mechanical dissociation ensured a single-cell suspension suitable for culture. Following dissociation, the sample underwent centrifugation at $700 \times g$ at 4 °C for 5 min, after which the supernatant was discarded, and the remaining cell pellet was resuspended in FGM. Fibroblast Growth Medium (FGM) was used to promote fibroblast attachment and proliferation. Subsequently, the resuspended cells were plated onto a 10 cm dish, with a media change performed within 24 h. Cells were cultured at 37 °C with 5% $CO_2$ and inspected daily to ensure uniform growth and absence of contamination.

## Mass spectrometry Imaging—histology

A total of 10 normal adult axilla, and 11 HS skin lesions, from both the right and left sides of patients, were excised and flash frozen in liquid nitrogen. All skin samples were cryosectioned into 20 µm sections and thaw mounted onto ITO or histology slides for MALDI and DESI imaging analysis.

## MALDI

Slides were coated with 10 mg/mL Diaminonaphthalene (DAN) with 0.1% Trifluoroacetic acid (TFA) using an HTX Imaging TM-sprayer for MALDI FT-ICR MSI analysis and imaged using a Bruker Solarix 12 T FT-ICR mass spectrometer in negative ion mode with a 40 µm raster width at 128 K. A laser spot size of ~10 µm was used to image bisphenols, while a spot size of ~30 µm was used to image phthalates. MALDI samples were analyzed using SCiLS Lab and uploaded to Metaboscape for analyte annotation.

## DESI

Slides were dried in a desiccator for 30 min prior to DESI analysis. DESI imaging was conducted using a Waters Cyclic-IMS mass spectrometer with a 50 µm raster width and 1.0 s scan speed. 80% acetonitrile was used as the electrospray solvent.

## Mass spectrometry imaging—ex vivo cultures

Eight CNTRL and eleven HS ex vivo cultures were rinsed three times in phosphate buffer solution (PBS) and fixed using 4% Paraformaldehyde in water before desiccation. Samples were then coated with 10 mg/mL Diaminonaphthalene with 0.1% Trifluoroacetic acid (TFA) in 50:50 acetonitrile:water (4 passes, 75 °C) using an HTX imaging TM-sprayer.

Samples were then imaged at 40 µm raster width using a Bruker Solarix FT-ICR mass spectrometer. Imaging data was compiled and compared using SCiLS Lab Pro v2024.

## Phosphoproteomics analysis

Nine samples of fibroblasts were cultured as follows: four HS skin lesions, three normal adult axilla, two amalgamations of foreskins (one control and one treated with p-ED cocktail as previously described). Proteins were harvested as done for western blot and stored at −80 °C before shipment to Poochon Scientific for mass spectrometry. Protein quantification was performed using BCA, and a total of 11 protein samples (50 µg each) were derived: four HS, three normal axilla controls, three technical replicates from the normal foreskin fibroblast control, and one normal fibroblast sample treated with the cocktail. The samples were digested with trypsin, and the trypsin-digested peptides (45 µg) were each uniquely labeled using the Tandem Mass Tag (TMT) multiplex set. An additional master mix, comprising of 5 µg from each of the 11 samples, was also prepared and TMT-labeled. The 12 labeled samples were then mixed and separated into two groups: 25% for the profiling of global proteins and 75% for enrichment of phosphopeptides, which was completed using the TiO2 enrichment kit. These groups were then each fractionated via reverse-phase ultra-high performance liquid chromatography (UHPLC), followed by liquid chromatography-tandem mass spectrometry (LC-MS/MS) analysis. The relative abundances of proteins was determined using TMT-tag based quantification and exported as a dataset.

In summary, 5038 proteins were identified across all 11 samples, and 730 proteins were phosphorylated, resulting in a total of 1601 phosphopeptides. The abundances of phosphopeptides were analyzed in R. Briefly, the dataset was cleaned, and any missing values were removed before performing quantile normalization. A Student's $t$ test was then conducted to filter for phosphopeptides with significantly different abundances between the HS and normal axillary control, identifying 650. The data were then converted to the logarithmic scale, and average log-fold changes were calculated for HS versus normal axillary and fibroblasts treated with the cocktail versus untreated. These were then fitted to a linear model, which was plotted along with the log-fold changes to produce a correlation plot.

## Bulk RNAseq

HS and CNTRL Fibroblasts were grown for experimental duration (7 days) in 6-well plates. Cells were treated with p-ED cocktail (20 nM), gamma secretase inhibitor, or vehicle control. TNFα was added to the media 48 h before RNA extraction. Bulk RNA extraction was performed using a QIAGEN RNeasy Mini Kit (QIAGEN 74104) per manufacturer's instructions. RNA sequencing was subsequently performed by Novogene. The relative expression (RPKM) of differentially expressed genes (defined as DESeq2 $p \leq 0.05$ and |log2FoldChange| $\geq 0.0$) was analyzed using R (4.4.0).

## Statistics

All experiments were performed in at least three individual instances, unless otherwise noted. To perform univariate statistical analysis, the Student's $t$-test was used. For multivariate analyses, the one-way ANOVA test was used. Detailed comparison method was labeled in figure legend. Statistical significance was defined based on $p < 0.05$ (*), 0.01 (**), 0.001 (***) or 0.0001 (****), either $p$ value or the defined results were marked in the figures. Data are represented as Mean ± SEM. No statistical method was used to predetermine sample size. No data were excluded from the analysis.

## Ethics statement

All experiments comply with all relevant ethical regulations. NIH Johns Hopkins IRB approved the protocols. Collection of CNTRL normal axillary samples under NA_00033375 "Skin biopsy study to determine gene expression signature of skin" and HS excisional samples under NA_00031269 "Use of Discarded Human Skin Surgery Tissue for Dermatology Research". Written informed consent was obtained from donors. Research in humans abided by the Declaration of Helsinki. Patient featured in Fig. 1a gave informed consent for image publication.

## Reporting summary

Further information on research design is available in the Nature Portfolio Reporting Summary linked to this article.

## Data availability

The bulk RNA sequencing data generated in this study have been deposited in the NCBI GEO database under accession code GSE307069. The mass spectrometry data are available via ProteomeXchange with identifier PXD068006. The scRNAseq data used in this study are available in the NCBI GEO database under accession codes GSE154773 and GSE175990. Source data are provided with this paper.

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

## Acknowledgements

Most importantly, we thank patients who elected to be included in this study; without their generosity, this work would be impossible. We thank M. White for his advice on experiments and the SKCCC Oncology Tissue and Imaging Services Core (Johns Hopkins School of Medicine) for tissue embedding and slide preparation services. We thank the Department of Dermatology's CTReP (Johns Hopkins School of Medicine) for equipment and infrastructure use. Research reported in this publication was supported by the Ina R. Drew and Howard J. Drew Innovation Fund, the National Institute of Arthritis and Musculoskeletal and Skin Diseases, part of the National Institutes of Health (NIH), under R01 ARO83822, T32 GM136577 and R56 ARO82660, the Maryland Stem Cell Research Fund Award 2022-MSCRFD-5917 to LAG, as well as the Daniel Nathans Scholar fund to LAG.

## Author contributions

K.L.W. and L.A.G. conceived the project. K.L.W. designed and performed majority of experiments and drafted the manuscript. B.B. assisted in processing patient tissue, consenting patients, and completing preliminary research to determine cocktail composition. J.R. assisted in cell culture experiments, maintenance, and western blot experiments/analysis, and performed screening work for inflammatory markers. W.A. performed mass spectrometry imaging experiments and analysis. H.B.M. assisted in cell culture maintenance and Supplementary RT-qPCR analysis. N.R.H. provided literature review. E.G.M. and N.S. extracted, maintained, and provided normal fetal fibroblasts for experiments. S.L. and A.L. maintained general laboratory equipment and reagents, and provided experimental guidance and technical feedback. L.C.Q., A.J., and A.W. consented normal axillary sample patients and provided the biopsies. C.S.K. gave essential feedback and technical guidance. A.V. assisted with proteomics experiments. J.C. performed excisional surgeries and C.C. and S.P.R. on the surgery team coordinated sample availability with B.B. K.K. reviewed the data and manuscript. N.A. provided critical feedback on experiments. M.K. ran the mass spectrometry imaging lab. L.A.G. supervised and provided feedback on experiments. All authors revised the manuscript.

## Competing interests

Research reported in this publication was supported by the Ina R. Drew and Howard J. Drew Innovation Fund, the National Institute of Arthritis and Musculoskeletal and Skin Diseases, part of the National Institutes of Health (NIH), under R01 ARO83822, T32 GM136577 and R56 ARO82660, the Maryland Stem Cell Research Fund Award 2022-MSCRFD-5917 to L.A.G., as well as the Daniel Nathans Scholar fund to L.A.G. The remaining authors declare no competing interests.
