## [Transparent Peer Review file · Nature Communications]

Plastic associated endocrine disruptors reduce Nicastrin protein and potentiate inflammation in Hidradenitis Suppurativa skin disease

Corresponding Author: Professor Luis Garza

Version 0:

Reviewer comments:

Reviewer #1

(Remarks to the Author)

The paper "Plastic associated endocrine disruptors reduce Nicastrin protein and potentiate inflammation in Hidradenitis Suppurativa skin disease" by Williams et al. describes a potential pathogenic mechanism accounting for the severity of Acne Inversa or Hidradenitis suppurativa in overweight patients.

The Authors made a considerable effort to associate the consumption of endocrine-disrupting chemicals contained in ultra-processed foods and susceptibility to sporadic hidradenitis suppurativa

The experiments are well designed, and the results are consistent, opening a new field of investigation for the impact of environmental factors on many disorders whose frequencies are rising in these times.

Major points to be discussed:

It is not clear to me throughout all the manuscript whether the Authors claim the consumption of UPF to be associated with susceptibility to the disease or, once the first pathogenic events have started to be a major factor related to disease severity. In this respect, The Authors stated that UPF consumption might cause "common acquired HS". Given that the heritability of HS is estimated at around 77%, how did the Authors choose the patients they analysed in this study?

A better characterisation of these patients is necessary. Do they have a family history of the disorder? Do they have genetic mutations that are associated with the disease? Can they be characterised in a specific latent class?

I'm convinced that a better description of the patients is needed in Table 1 and, if the patients belong to a specific latent class, a comparison with other specific classes would significantly improve the results.

The Authors never explained how the p-EDs specifically knocked down NCSTN protein expression in fibroblasts. A part reporting that bisphenol A can "bind and impact GS", they haven't looked at the molecular mechanism.

p-EDs impact protein expression or gene transcription? As Authors have access to cells isolated from patients, they should be able to experiment with whether the p-EDs are able block the transcription of the gene or impact the maturation of the g-secretase and thus the half-life of the subunits.

Also GS has 4 essential subunits and the Authors should investigate whether p-EDs impact the quantity of other GS components.

Throughout all the paper, the Authors have looked in to the ability of p-EDs to potentiate fibroblasts responses to TNF-a. Today we know several cytokines are involved in severity of HS such as IL-17, IL-12/23 and for some patients IL-1b. All the patients from this study have elevated levels of TNF-a. Do they respond to the therapy against TNF-a?

May the Authors show whether their findings could be reproduced treating the fibroblasts with IL-17 or IL-1b?

NCSTN expression is enhanced by TNF-a treatment. Do the Authors know whether this is common in all cells or specific to fibroblasts? Does the TNF-a stimulation increase the transcription of the gene or the half-life of the protein?

Is TNF-a stimulation able to increase the expression of the other subunits of the g-secretase?

An over-expression experiment would help to clarify whether the NCSTN alone is able to increase NF-Kb translocation to

the nucleus or whether the two events are only associated, as for now it is not clear

The Authors jump to the conclusion that loss of NCSTN, both genetic and after exposure to EDCs, impact mainly fibroblasts and their capacity to secrete pro-inflammatory cytokines causing HS lesions. In Authors opinion this is supported by scRNA-seq studies performed in skin of HS patients that confirmed a major pro-inflammatory role of dermal fibroblasts. Even if fibroblasts are probably very important in sustaining the inflammatory environment the Authors never explained 1) why the specific localization of lesions in HS, 2) why the earliest pathogenic changes take place around the hair follicles. Are there particular factors secreted by fibroblasts that specifically drive hair follicle cells proliferation? Pointing to the proteins secreted by fibroblasts that can specifically lead to the proliferation of hair follicle cells and disruption of the hair follicle niche would open new targets for treating the disease.

The authors made a great effort in characterizing proinflammatory cytokines and chemokines secreted by fibroblasts due to an increased translocation of NF-Kb in their nucleus. In Figure 1, a better characterisation of the immune infiltrate would substantiate Authors claim. As all the patients presented a reduced nicastrin expression in IF experiment, I would expect their fibroblasts to secrete IL-8 and then a massive infiltration of neutrophils should be present.

The Authors hypothesis fail to explain one of the major features of HS: the formation of fistula, tunnels /tendrils mainly composed by keratinocytes. In fact, whilst the production of inflammatory chemokines and cytokines is common with several genodermatosis, today it is becoming increasingly evident that the formation of tunnels in the skin of the patients is the main pathogenic feature of HS. May the Authors explain the role of fibroblasts and the lack of NCSTN expression for this feature

Minor points

- Figure 1 E: Images from HC and Patients do not seem to be recorded at the same magnification
- Figure 1 G : there seems to be lot of variation in protein load among wells that can invalidate the WB result
- Figure 2C : TNF- α promote expression of NCSTN in cells scRNA treated cells, whilst this is not observed in siRNA NCSTN treated cells – is maybe a result dependent on the siRNA used? May the authors obtain the same results as those of another siRNA against NCSTN?
- Figure 2C vs 2E : siRNA efficiency seem different in NCSTN KD from one experiment to another
- Figure 3 E: It is not clear to me what the Authors want to show with the WB of HES-1
- Figure 3 M-N : TNF- α treatment of fibroblasts transfected with scRNA failed to enhance IL-8 expression: maybe a problem with the scRNA?
- Figure 4 H: why at 20 μ M p-EDs the NCSTN expression is lost, but IL-8 secretion is not increased as expected?

Reviewer #2

(Remarks to the Author)
NCOMMS-24-67119T

In this manuscript, the authors identified a loss of NCSTN in dermal fibroblasts of acquired HS patients. This event potentiates inflammatory responses in fibroblasts. They carried out mass spectrometry imaging that revealed that p-ED are significantly enriched in HS patients' tissues. This study was well designed and collected important information of HS patients to figure out molecular mechanisms. However, the authors should address very carefully the following points:

- (1) The authors used two different types of MS imaging; MALDI and DESI. Methods did not describe the detailed information. Choice of matrix compound in MALDI? The authors should show the original mass spectra to compose the tissue images. Figure 4 showed differences in bisphenol and phthalates between the control and HS. Did the authors check if ATP is detected in the same samples? Such information benefits a quality control of the frozen samples.
- (2) The authors need to show the original MS spectrum to visualize bisphenol and phthalates in Supplementary information.
- (3) The authors showed increases in several cytokines that cause inflammatory responses in HS skin. Can the authors visualize elevation of any metabolites which might be contained in inflammatory cells?
- (4) Phthalates are known to directly bind to glycogen-debranching enzyme to perturb glucose metabolism. Did the authors recognize any alterations in glycolytic metabolites, if any.
- (5) Ex vivo MSI: Under conditions of culture systems, the presence of phosphate in the culture medium might disturb ionization of metabolites. How did the authors prepare ex vivo cell culture for MSI? This reviewer was unable to check the detailed information (How to fix cells, Composition of the buffer, etc) for experimental setup. Provide these lines of information.

Reviewer #4

(Remarks to the Author)
In this manuscript the authors provide evidence that the acquired Hidradenitis Suppurativa (HS) inflammatory skin condition may be associated with exposure to environmental endocrine disrupting chemicals (EDCs). Notably they proposed that

plastic-associated EDCs (p-Eds) that may be present in ultraprocessed foods (UPFs) contribute to the development of HS since they and others had previously linked a diet containing UPFs with HS. Correlative evidence exists that suggests association of EDCs with HS since it is more prevalent in females, flares are associated with the menstrual cycle and incidence is highest in populations with the highest consumption of UPFs. The main thrust of the article is the linkage the authors establish between Nicastrin loss in dermal fibroblasts of acquired HS patients and a corresponding connection between Nicastrin loss and inflammation in dermal fibroblasts. The authors found that p-EDs are also enriched in dermal fibroblasts from HS patients compared with controls. Importantly, the current manuscript found that acquired HS is linked with hereditary HS via impairment of Nicastrin function.

Establishing a causal link between EDCs and HS would add another link between UPFs, EDCs and adverse health outcomes and could inform future preventive measures aimed at reducing or eliminating HS. As the authors correctly point out, it is difficult to distinguish causal factors from those that are secondary or HS-associated themselves. For example, the authors noted that 80-100% of study populations contain bisphenol A (BPA) in multiple tissues but fail to note that this is also the frequency of BPA detection in non-affected populations. It is also unclear why BPA accumulated in dermal fibroblasts of HS patients compared with the controls, which presumably represent the unaffected population that has similar frequency of BPA detection. It is also unclear why p-EDs accumulated in dermal fibroblasts compared with healthy axillary fibroblasts from the same individuals.

As the authors note starting on line 328, this study raises important questions for future study. The authors stated on l 324 that they have identified the molecular mechanism of p-EDS causing HS-like phenotypes, *in vitro*. I do not believe that this has been demonstrated adequately. There is a correlation between p-EDs levels and HS and this continues in dermal fibroblasts cultured *ex vivo*, but the manuscript falls short of demonstrating causation. Perhaps a better presentation of critical evidence would be more persuasive but the manuscript in its current form does not.

Important issues:

Overall, I find the data to be extensive and of apparent high quality. However, one is left with the impression that there is a tremendous amount of data that is hard to interpret because it is crammed into a very small space. This is compounded by inadequate discussion of what these extensive data show. I believe that the authors should strive to make the data depicted readily accessible to the readership of Nature Communications and keep only the essential panels from each figure in the main results and move the remainder to Extended Data.

Another important issue with respect to the figures is the extremely small size of the panels in each. I do not understand why the authors did not make the figures themselves full page with the legends on an additional page for the purposes of review. This makes it extremely difficult to interpret the figures. As noted above, it could be beneficial to reduce the number of panels in each figure so that they can be presented at a resolution that is accessible to the reader.

Figure 2f is particularly important to the author's model since this reflects siRNA knockdown of Nicastrin and its effect on the expression of various genes that are presumably changed in HS. While I appreciate the compact nature of this figure and how clearly the differences between fibroblasts and keratinocytes are shown, the figure really fails to make the point that particular genes whose expression is altered in HS are similarly affected in the siRNA-mediated knockdown. This would be much better shown in a table that relates which genes are the most affected in HS and their corresponding change in the siRNA knockdown.

What is being merged in Figure 4k? This is not noted in the legend or in the figure. The lettering throughout the figure panels is too small in many cases to identify what is being depicted.

Extended Data figure 5i is said to show a comparison of bulk RNAseq from normal dermal fibroblasts treated with the p-ED mixture vs. normal dermal fibroblasts treated with a gamma secretase inhibitor. As noted above, the panel is uninterpretable as presented, making it difficult to ascertain the validity of the authors' assertions.

Other comments to be addressed:

l 39-41 – It is unclear why the authors switched terminology to plasticizers here since bisphenols are not usually considered to be plasticizers

The Western Blots in Fig 1g showing are not terribly convincing with respect to the effects of TNF on the cultured fibroblasts. The immunofluorescence panels are very difficult to evaluate. One hopes that these will be large enough in the final version of the paper for the reader to distinguish the features being discussed. I cannot and can only take the descriptions by the authors at face value. Perhaps the quantitation of the panels could be shown in the main figures and the immunofluorescence shown with greater magnification and accessibility similar to Extended Data Figure 2?

Version 1:

Reviewer comments:

Reviewer #1

(Remarks to the Author)

Dear Editor,

The Authors have responded to most of the main concerns raised in the first review.
The manuscript has been greatly improved by experiments added by the Authors and a better description of patients under study

I still have some questions about the author's replies.

I understand the Authors have restricted information on the genetics of HS in the patients analysed, as family data are unavailable for most patients -still, adding available data has improved the characterisation of the cohort under study.

I have an issue regarding the choice of healthy controls and patients in this study.

HS affects mainly African American women and 90% of the patients are actually African American, why the Authors used other populations (mainly Caucasians and Asians) as controls?

Most of the healthy controls are overweight – do they consume ultra-processed foods as well, or do they have a completely different diet from HS patients?

Do the Authors have any information regarding diet preferences in the two cohorts?

How do the Authors explain that TNF α plays a major role in CXCL8's increased expression by fibroblasts in their patients, and yet many of them still failed to respond to Humira?

Is it worth to add few lines in the discussion section

For most of the experiments 8 HS were used and there are 12 patients enrolled in this study. May the Authors point which patients were actually analysed?

Figure 1 of the paper and rebuttal letter: Supplemental Fig. 7

I am concerned about Western blot and IF analyses on NCSTN expression. The antibody used in this study can detect both the mature and immature forms of the NCSTN protein.

Did the Authors analyze the mature and immature forms of the protein or they focus on just one form?

Actually, most of the figures throughout the manuscript (especially siRNA experiments) showed just one main band for the protein.

IF experiments cannot differentiate between the 2 forms, but WB gels should be shown to help readers understand which form was visible and analysed by the Authors.

For the supplemental Figure 7 both mature and immature forms of the protein should be analysed to understand the role of a defect in the maturation process of the protein.

Still, it is already known that a lack of a single subunit of the gamma-secretase complex may increase the degradation of g-secretase subunits.

Given the new results shown by the Authors that stated that : “ GS subunit mRNA are similarly affected by p-ED exposure in the presence of TNF α , so the effect is at least partially transcriptionally regulated and interestingly inhibiting multiple GS subunits” how can they be sure that p-ED exposure affects the transcription of an unspecified endonuclease able to specifically degrade NCSTN and do not affect the expression of other g-secretase subunits that cause an increased instability of mature NCSTN, thus reducing the levels of the total protein?

The Authors never provided an experimental link between decreased NCSTN expression and pNF- κ B nuclear translocation. As the authors focused only on NCSTN expression, it is unclear whether this could be related to the γ -secretase complex activity or NCSTN activity independent of other γ -secretase subunits.

It is unclear whether the Authors support the hypothesis that HS is primarily an autoinflammatory keratinisation disease, where the first insult is an auto-inflammatory response or is a disorder driven by the occlusion of the hair follicle.

In any case, where will the authors place an elevated consumption of p-ED in the pathophysiology of the disease?

Is an elevated consumption of processed food associated with the disease severity or may it cause the disease itself?

Can you please clarify this point in the discussion section?

I want to thank the Authors for addressing the remaining points raised in the first review of this manuscript.

Reviewer #2

(Remarks to the Author)

The authors replied satisfactorily to this reviewer by supplementing additional experimental data.

Reviewer #4

(Remarks to the Author)

The authors have responded to my critiques in a satisfactory way. I am still not convinced that they have established a causal relationship between plasma bisphenols and HS, but the data presented here are a valuable addition to the literature.

Version 2:

Reviewer comments:

Reviewer #1

(Remarks to the Author)

The authors have adequately responded to all the points of concern I have raised. I'm quite convinced that consumption of ultra-processed foods could be associated with the severity of the disease and this paper was able to disclose a link between the two, even with several limitations that the Authors outlined in the discussion. I'm sure that this will elicit the research for the environmental causes of this cumbersome disorder, even if lot of work remains to elucidate the physiological mechanisms disrupted by the consumption of UPF in the skin.

Title: Plastic associated endocrine disruptors reduce Nicastrin protein and potentiate inflammation in Hidradenitis Suppurativa skin disease

We sincerely thank the reviewers and editor for their time and thoughtful, constructive advice. We are grateful for their positive feedback on the article:

- “The experiments are well designed, and the results are consistent, opening a new field of investigation for the impact of environmental factors on many disorders whose frequencies are rising in these times.”
- “This study was well designed and collected important information of HS patients to figure out molecular mechanisms.”
- “Overall, I find the data to be extensive and of apparent high quality.”

Below, we have copied the reviewers’ comments verbatim and provided detailed, point-by-point responses, incorporating new experiments and textual revisions accordingly. We have spent the last 5 months carefully addressing each comment, revising the figures, adding new data, and supplementing our discussion. Furthermore, we have restructured some of our supplemental figures to better incorporate and represent data for the reviewers.

We are grateful for the reviewers’ efforts in highlighting areas for improvement and believe that these revisions have significantly strengthened the clarity and impact of our manuscript. Please find our detailed responses to each comment below.

REVIEWER COMMENTS

Reviewer #1 (Remarks to the Author):

The paper “Plastic associated endocrine disruptors reduce Nicastrin protein and potentiate inflammation in Hidradenitis Suppurativa skin disease” by Williams et al. describes a potential pathogenic mechanism accounting for the severity of Acne Inversa or Hidradenitis suppurativa in overweight patients.

The Authors made a considerable effort to associate the consumption of endocrine-disrupting chemicals contained in ultra-processed foods and susceptibility to sporadic hidradenitis suppurativa

The experiments are well designed, and the results are consistent, opening a new field of investigation for the impact of environmental factors on many disorders whose frequencies are rising in these times.

We appreciate Reviewer 1’s positive feedback on the overall impact and data in this manuscript.

Major points to be discussed:

It is not clear to me throughout all the manuscript whether the Authors claim the consumption of UPF to be associated with susceptibility to the disease or, once the first pathogenic events have started to be a major factor related to disease severity. In this respect, The Authors stated that UPF consumption might cause “common acquired HS”. Given that the heritability of HS is estimated at around 77%, how did the Authors choose the patients they analysed in this study?

The reviewer raises several important areas for improvement we have implemented in the present revised manuscript. Firstly, we have refined our terms used in the manuscript to be *monogenic* HS vs *sporadic* HS. We very much agree with the author’s point that even sporadic HS has a large genetic component, and we only mean to differentiate the greater penetrance effect of Gamma Secretase mutations in monogenic HS.

Secondly, we also agree with the reviewer’s insightful point that our work is fully compatible with a model where sporadic HS arises from an exposure plus genetic susceptibility that strengthens the effect of that exposure. This is very logical since UPFs use is much more widespread than HS. We have added these very important points to our discussion.

To respond to the reviewer’s question, our patient samples come from excisional tissue, which will by necessity be later stage (Hurley II/III) HS. Any patient receiving excisional surgery who met IRB criteria and consented to this research study was included.

A better characterisation of these patients is necessary. Do they have a family history of the disorder? Do they have genetic mutations that are associated with the disease? Can they be characterised in a specific latent class?

I’m convinced that a better description of the patients is needed in Table 1 and, if the patients belong to a specific latent class, a comparison with other specific classes would significantly improve the results.

The reviewer raises an important point for better description of patients studied in the present manuscript. Patients are not routinely tested for genetic mutations associated with HS, but it is apparently from our immunofluorescence data (Fig1c/d) that all patients have intact NCSTN protein in their non-fibroblast cells, especially keratinocytes, suggesting they are not NCSTN missense mutation patients.

We agree a better description of patient characteristics in Table 1 can be helpful in more completely describing our cohort. We’ve added new data regarding current and failed biologic therapies, years since diagnosis, and family history.

As a practical method to try to identify any latent classes, we performed a FAMD analysis on our HS and NAX (normal axillary control) patients, testing quantitative variables fibroblast NCSTN score (relative fluorescent intensity of tissue stain), age, and BMI and qualitative variables Hurley Stage, Surgical Site, Race, and Sex. In this analysis, higher NCSTN stain was a stronger predictor of NAX identity than BMI was of HS origin. Individuals cluster on a dimensional analysis graph within their Hurley stage; high NCSTN stain is a strong variable predictive of “NAX” of an individual in the analysis. Our patients cluster strongly by Hurley stage or NAX status. We agree that latent classes are certainly likely in HS, but are not discernable given our current data; more work is important in this domain. We have added a point in discussion that the field must do more to identify different endotypes of HS and disclosed this important caveat for our data.

Table 1. Demographic information of study participants

PUB ID	Site of Surgery	Sex	Age	Race	BMI	Hurley Stage	Family History of HS	Time since symptoms began	Current Biologic Therapy	Failed Biologic Therapy
HS1	left axilla	F	46	White	39.89	3	NA	5 years	Infliximab	Humira
HS2	lower abdomen, mons pubis, bilateral groin	F	39	Asian	35.82	3	NA	>10 years	no	Humira
HS3	bilateral buttocks	F	67	African American	40.3	3	NA	>40 years	no	Humira
HS4	groin, perineum, lower abdomen, mons	F	26	Hispanic	38.69	3	NA	>20 years	no	Humira, infliximab
HS5	right axilla	F	22	White	27.46	2	NA	n/a	Cosentyx	n/a
HS6	lower abdomen, mons pubis, bilateral groin	F	28	White	42.4	not assigned	NA	>5 years	Humira	n/a
HS7	bilateral axilla	F	47	African American	35.61	3	NA	>10 years	Humira	n/a
HS8	abdomen, bilateral thighs, groin, mons	F	36	African American	37.31	3	NA	15 years	Humira	Humira
HS9	bilateral axilla	F	44	African American	36.64	3	Mother	>20 years	Cosentyx	Humira, infliximab
HS10	bilateral axilla	F	24	African American	37.68	3	Aunt	>10 years	Humira	n/a
HS11	bilateral buttocks, sacrum	M	43	African American	24.4	3	NA	>18 years	no	Humira
HS12	lower abdomen, left labia groin and thigh	F	48	African American	35.61	3	NA	n/a	Humira	n/a
NAX1	axilla	F	45	African American	39.71	NA	NA	NA	NA	NA
NAX2	axilla	F	28	Asian	not available	NA	NA	NA	NA	NA
NAX3	axilla	F	22	White	20	NA	NA	NA	NA	NA
NAX4	axilla	M	68	White	29.15	NA	NA	NA	NA	NA
NAX5	axilla	F	21	White/Hispanic	29.3	NA	NA	NA	NA	NA
NAX6	axilla	F	21	Asian	18.7	NA	NA	NA	NA	NA
NAX7	axilla	M	35	Asian	28.2	NA	NA	NA	NA	NA
NAX8	axilla	F	24	Asian	not available	NA	NA	NA	NA	NA
NAX9	axilla	F	24	Asian	24.2	NA	NA	NA	NA	NA

The Authors never explained how the p-EDs specifically knocked down NCSTN protein expression in fibroblasts. A part reporting that bisphenol A can “bind and impact GS”, they haven’t looked at the molecular mechanism. p-EDs impact protein expression or gene transcription? As Authors have access to cells isolated from patients, they should be able to experiment with whether the p-EDs are able block the transcription of the gene or impact the maturation of the g-secretase and thus the half-life of the subunits. Also GS has 4 essential subunits and the Authors should investigate whether p-EDs impact the quantity of other GS components.

We agree with the reviewer that how p-EDs knock down NCSTN protein expression is of particular interest. We have added new data in this regard to the manuscript to add more experimental information to address the reviewer's comment. We completed additional RT-qPCR experiments that show other GS subunit mRNA are similarly affected by p-ED exposure in the presence of TNF α , so the effect is at least partially transcriptionally regulated and interestingly inhibiting multiple GS subunits. (see below; added to **Supplemental Fig. 7a**). Of note, our transcriptional data shows a loss of NCSTN at the mRNA level with the addition of only p-EDs, while our protein results only show loss of NCSTN in the addition of both p-EDs and TNF α . This suggests control of NCSTN protein at both the transcriptional and post-translational level.

To address a role for p-EDs in modulating NCSTN at the translational or post-translational level, we also conducted further experiments to add to this revision. To further characterize NCSTN protein loss with p-ED treatment, we added a proteasome inhibitor (MG 132) and an inhibitor of translation (puromycin) to the experiment to determine how NCSTN protein levels are modified. Puromycin, but not MG132, at least partially rescued NCSTN protein in the presence of p-EDs and TNF α (See below; added to **Supplemental Fig. 7g**). These results suggest that perhaps NCSTN is not degraded by the proteasome given the negative effect of MG 132. They suggest instead that p-EDs induce the translation of a protease; NCSTN could be degraded by an enzyme that is translationally controlled, such as a protease. Thus in the absence of translation of this protease, NCSTN is somewhat rescued. The limitation of these results is that they are using conventional concentrations of these inhibitors, and timing and concentration changes might have different results. Nevertheless, these data suggest both transcriptional and translational effects of p-EDs on NCSTN protein abundance.

NCSTN - combined westerns

Representative WB

Throughout all the paper, the Authors have looked into the ability of p-EDs to potentiate fibroblasts responses to TNF-a. Today we know several cytokines are involved in severity of HS such as IL-17, IL-12/23 and for some patients IL-1b. All the patients from this study have elevated levels of TNF-a. Do they respond to the therapy against TNF-a?

We have added anti-TNFa therapy status in our demographic patient Table 1. All of our patients were either currently on, or previously attempted, Humira therapy, to variable success. Exactly half (6/12) of our patient cohort stopp-ED responding to Humira therapy at some point prior to surgery. We have added this information to the manuscript.

May the Authors show whether their findings could be reproduced treating the fibroblasts with IL-17 or IL-1b?

We agree this is an important question, especially with the success of Cosentyx and Bizekizumab in HS. In the interim months since submission, we have conducted experiments using IL-17 to see if it mirrors TNFa treatment in our system.

We tested the effects of IL-17 by flow cytometry as below. We found IL-17 does not induce increased NCSTN protein as in TNFa treatment, which is interesting. This points to some unique biology between IL-17 and TNF.

However, IL-17 Together with TNFa induces very unique amounts of IL-8 in dermal fibroblasts. This suggests some important synergy. Further projects exploring how IL-17 signaling potentiates HS and IL8 would be very interesting and augment our study. We have included this important future direction in our discussion.

NCSTN expression is enhanced by TNF- α treatment. Do the Authors know whether this is common in all cells or specific to fibroblasts? Does the TNF- α stimulation increase the transcription of the gene or the half-life of the protein?

We thank the reviewer for pointing this out specifically; while not the major finding of this manuscript, we think it is extremely interesting. We show in **Supplemental Figure 7a** TNF α increases NCSTN mRNA, as well as other GS member mRNA, so it is at least a partially transcriptionally regulated process. The reviewer's insightful point highlights an important biological connection between inflammation and gamma secretase function that is underappreciated.

This effect does not exist in keratinocytes (NHEKs, specifically), at least not reflected at the protein level, which is represented in Figure 4e.

Is TNF- α stimulation able to increase the expression of the other subunits of the g-secretase?

We have in the interim confirmed via RT-qPCR that TNF α stimulation does increase the transcription of other subunits of gamma secretase. Specifically, APH1B, PSEN2, and PSENEN all increase substantially with TNF α stimulation, as well as NCSTN. See below (also shown above in review, is now in **Supplemental Fig 7a**). In the context of p-EDs common effect on inhibiting multiple GS subunits, the common effect of TNF α on stimulating multiple GS subunits is likely especially relevant.

An over-expression experiment would help to clarify whether the NCSTN alone is able to increase NF-Kb translocation to the nucleus or whether the two events are only associated, as for now it is not clear

According to our hypothesis, overexpression of NCSTN would prevent or decrease pNF-kB nuclear translocation. Loss of NCSTN promotes pNF-kB nuclear translocation (see below). Nevertheless, the reviewer raises an important future direction of Gain-of-function experiments on NCSTN. We have added this to the discussion.

The Authors jump to the conclusion that loss of NCSTN, both genetic and after exposure to EDCs, impact mainly fibroblasts and their capacity to secrete pro-inflammatory cytokines causing HS lesions. In Authors opinion this is supported by scRNA-seq studies performed in skin of HS patients that confirmed a major pro-inflammatory role of dermal fibroblasts Even if fibroblasts are probably very important in sustaining the inflammatory environment

the Authors never explained 1) why the specific localization of lesions in HS, 2) why the earliest pathogenic changes take place around the hair follicles.

We thank the reviewer for this insightful question. The most important point is that we have added new comments to the discussion that our focus on fibroblasts, GS, and p-EDs do not preclude the likely importance of parallel pathways that promote HS disease. The work of all investigators in the HS space is very important to fully understand this highly multifactorial disease.

1. We are also very interested in why HS lesions occur in primarily intertriginous areas. We have several overlapping hypotheses, several which are under active investigation in our laboratory. Of note, we hypothesize sweat exposure is a major driver. P-EDs are lipid-soluble molecules, and eccrine and apocrine glands could be secreting p-EDs in sweat. We also wonder if the occlusion of intertriginous areas induces more p-ED reabsorption into a new damaging tissue depot. This long and repeated exposure could be an explanation for the localization of HS. We have further explored this question in two publications (PMID: 38758298 & 39481530).
2. The known hair follicle association with HS is also of interest to us. While we did not present data directly related to hair follicles in this paper, we believe there are several hypotheses that may tie into our findings and this known phenomenon. For example, occlusion of hair follicles is often thought to be a primary driver of beginning disease (PMID: 30619323). Our findings in this work suggest an initial inflammatory insult (modeled with TNF α treatment) begins the process of NCSTN loss and inflammatory potentiation. Also, we wonder if GS biology connects to morphogenic events in hair follicle subpopulations to explain a HF association with HS. This is supported by some old work from the lab of Raphael Kopan.

Are there particular factors secreted by fibroblasts that specifically drive hair follicle cells proliferation? Pointing to the proteins secreted by fibroblasts that can specifically lead to the proliferation of hair follicle cells and disruption of the hair follicle niche would open new targets for treating the disease.

The reviewer brings up an exciting area of future study. As others, we noted several findings in published scRNAseq datasets that may provide additional context. We identified versican positive – likely dermal papillae or hair follicle associated – fibroblasts (VCAN+, MMP3+, MMP1+) to be the main sources of both CXCL13 and MMP3 in HS patients. In the scRNAseq data from GSE175990, this population of fibroblasts has a 4.869 log fold change (adjP = 2.77E-221) in CXCL13 expression in HS samples compared to a healthy control (HC). This population of fibroblasts also has CXCL13 expression levels that are 13 fold and 5 fold higher than the next most abundant CXCL13 producer, T cells, in GSE154775 and GSE175990 respectively.

These findings were published in an abstract (ISID 2023 1622) ; we have mentioned this in the discussion.

The authors made a great effort in characterizing proinflammatory cytokines and chemokines secreted by fibroblasts due to an increased translocation of NF-Kb in their nucleus.

In Figure 1, a better characterisation of the immune infiltrate would substantiate Authors claim. As all the patients presented a reduced nicastrin expression in IF experiment, I would expect their fibroblasts to secrete IL-8 and then a massive infiltration of neutrophils should be present.

We thank the reviewer for this suggestion and agree it would improve the manuscript. A significant limiting factor is our study is composed of late-stage HS patients. In these patients, inflammatory infiltrate appears to be largely dominated by lymphocytes, which matches much of published literature indicating B cells have high importance of late disease. In contrast, previously published literature shows neutrophils as predominant contributors to early HS lesions alongside macrophages (PMID: 21929531 & 26436522).

Regardless, we have performed additional IHC staining of HS tissue and see neutrophils recruited to the dermis as expected. (now **Supplemental Fig. 4m**)

The Authors hypothesis fail to explain one of the major features of HS: the formation of fistula, tunnels /tendrils mainly composed by keratinocytes.

In fact, whilst the production of inflammatory chemokines and cytokines is common with several genordematosis, today it is becoming increasingly evident that the formation of tunnels in the skin of the patients is the main pathogenic feature of HS

May the Authors explain the role of fibroblasts and the lack of NCSTN expression for this feature

Tunnel formation is a fascinating aspect of HS and we agree deserves further discussion.

We agree tunnel morphogenesis is a vital and fascinating future direction. We added a discussion of additional future directions in this regard in the discussion.

Minor points

Figure 1 E: Images from HC and Patients do not seem to be recorded at the same magnification

These are recorded at the same magnification; there is some variation amongst fibroblast sizes. We double checked our scale bars and magnification.

Figure 1 G : there seems to be lot of variation in protein load among wells that can invalidate the WB result

We normalize all our western blots to their respective B-Actin loading control, but agree presentation could be improved. We have moved the blot from our main figure to supplemental data. Since we have included confirmation by IF, and repeated the western blot several times we are confident in the data.

Figure 2C : TNF- α promote expression of NCSTN in cells scRNA treated cells, whilst this is not observed in siRNA NCSTN treated cells – is maybe a result dependent on the siRNA used? May the authors obtain the same results as those of another siRNA against NCSTN? We thank the reviewer for this point - we errantly did not include ThermoFisher siNCSTN, which were also used for some experiments in the methods. We have updated the methods.

Figure 2C vs 2E : siRNA efficiency seem different in NCSTN KD from one experiment to another

While there is some minor variability between experiments on NCSTN ND efficiency (possibly due to different companies' specific siRNAs), all but 1 replicate have > 80% reduction in NCSTN protein abundance. We believe this, in combination with many of our IF and RT-qPCR experiments, is an accurate representation of the biology of significant loss in NCSTN protein.

Figure 3 E: It is not clear to me what the Authors want to show with the WB of HES-1
Thank you for noticing; we mistakenly included the HES-1 result and have removed it, as it is not relevant to our findings.

Figure 3 M-N : TNF- α treatment of fibroblasts transfected with scRNA failed to enhance IL-8 expression: maybe a problem with the scRNA?

TNF α does stimulate IL8 in scRNA treated cells; just at a much lower level. See quantification in 3n, as well as RT-qPCRs, westerns, and IF throughout figure 3.

Figure 4 H: why at 20 μ M p-EDs the NCSTN expression is lost, but IL-8 secretion is not increased as expected?

This is a fascinating phenomenon and we do not have an exact explanation for it. In several subsequent experiments performed in the interim, the same pattern persists.

Reviewer #2 (Remarks to the Author):

NCOMMS-24-67119T

In this manuscript, the authors identified a loss of NCSTN in dermal fibroblasts of acquired HS patients. This event potentiates inflammatory responses in fibroblasts. They carried out mass spectrometry imaging that revealed that p-ED are significantly enriched in HS patients' tissues. This study was well designed and collected important information of HS patients to figure out molecular mechanisms. However, the authors should address very carefully the following points:

(1) The authors used two different types of MS imaging; MALDI and DESI. Methods did not describe the detailed information. Choice of matrix compound in MALDI? The authors should show the original mass spectra to compose the tissue images. Figure 4 showed differences in bisphenol and phthalates between the control and HS. Did the authors check if ATP is detected in the same samples? Such information benefits a quality control of the frozen samples.

We would like to thank the reviewer for their insightful feedback. The matrix used was 10 mg/mL Diaminonaphthalene (DAN), referenced on line 682. We will also add that the matrix included 0.1% Trifluoroacetic acid (TFA) on line 683. The reviewer is correct in that raw spectra of detected analytes should be included in the manuscript. We have added the spectra for the parent ions from the overall average spectrum for all of the HS samples, as well as the overall average spectrum for all of the CNTRL samples in **Supplemental Fig 5**.

While we didn't optimize the protocol for detecting ATP and ADP, ADP was observed to be more intense in CNTRL samples when compared to HS samples. In addition, NADH was more intense in CNTRL samples when compared to HS as well. ATP was detected with similar intensities in both sets of samples. These data are also now available in **Supplemental Fig. 5**.

(2) The authors need to show the original MS spectrum to visualize bisphenol and phthalates in Supplementary information.

We appreciate the reviewer's insightful feedback. We have included the parent ions from the overall average spectra of HS and CNTRL tissues that were used to visualize these analytes (see above).

(3) The authors showed increases in several cytokines that cause inflammatory responses in HS skin. Can the authors visualize elevation of any metabolites which might be contained in inflammatory cells?

We agree this is an excellent way to further support our hypothesis. We have detected arachidonic acid at much higher levels in the HS samples than CNTRL. These data are also now available in **Supplemental Fig. 5**.

(4) Phthalates are known to directly bind to glycogen-debranching enzyme to perturb glucose metabolism. Did the authors recognize any alterations in glycolytic metabolites, if any.

While our protocol was not optimized for glycolytic metabolites, several glycolytic metabolite candidates were observed. However, several of these species have the same molecular formula, so further LC-MS based experiments would need to be conducted to confirm their identities. Parent ions for metabolite candidates G3P or DHAP, glucose, and pyruvate were all observed, with increased intensity for G3P/DHAP and pyruvate in HS samples when compared to NAX samples. These data are also now available in **Supplemental Fig. 5**.

(5) Ex vivo MSI: Under conditions of culture systems, the presence of phosphate in the culture medium might disturb ionization of metabolites. How did the authors prepare ex vivo cell culture for MSI? This reviewer was unable to check the detailed information (How

to fix cells, Composition of the buffer, etc) for experimental setup. Provide these lines of information.

Thank you for identifying this omission. We agree with the reviewer that the *ex vivo* cell culture procedure was not described in great detail, and have modified the methods to include the following:

Eight NAX and eleven HS *ex vivo* cultures were rinsed three times in phosphate buffer solution (PBS) and fixed using 4% Paraformaldehyde in water before desiccation. Samples were then coated with 10 mg/mL Diaminonaphthalene with 0.1% Trifluoroacetic acid (TFA) in 50:50 acetonitrile:water (4 passes, 75 degrees C) using an HTX imaging TM-sprayer. Samples were then imaged at 40 um raster width using a Bruker Solarix FT-ICR mass spectrometer. Imaging data was compiled and compared using SCiLS Lab Pro v2024.

Reviewer #4 (Remarks to the Author):

In this manuscript the authors provide evidence that the acquired Hiradenitis Suppurativa (HS) inflammatory skin condition may be associated with exposure to environmental endocrine disrupting chemicals (EDCs). Notably they proposed that plastic-associated EDCs (p-Eds) that may be present in ultraprocessed foods (UPFs) contribute to the development of HS since they and others had previously linked a diet containing UPFs with HS. Correlative evidence exists that suggests association of EDCs with HS since it is more prevalent in females, flares are associated with the menstrual cycle and incidence is highest in populations with the highest consumption of UPFs. The main thrust of the article is the linkage the authors establish between Nicastrin loss in dermal fibroblasts of acquired HS patients and a corresponding connection between Nicastrin loss and inflammation in dermal fibroblasts. The authors found that p-EDs are also enriched in dermal fibroblasts from HS patients compared with controls. Importantly, the current manuscript found that acquired HS is linked with hereditary HS via impairment of Nicastrin function.

Establishing a causal link between EDCs and HS would add another link between UPFs, EDCs and adverse health outcomes and could inform future preventive measures aimed at reducing or eliminating HS. As the authors correctly point out, it is difficult to distinguish causal factors from those that are secondary or HS-associated themselves. For example, the authors noted that 80-100% of study populations contain bisphenol A (BPA) in multiple tissues but fail to note that this is also the frequency of BPA detection in non-affected populations. It is also unclear why BPA accumulated in dermal fibroblasts of HS patients compared with the controls, which presumably represent the unaffected population that has similar frequency of BPA detection. It is also unclear why p-EDs accumulated in dermal fibroblasts compared with healthy axillary fibroblasts from the same individuals.

We thank the reviewer for their insightful question; we also find it is interesting p-ED accumulation is specific to dermal fibroblasts. We have added to the discussion that HS is

likely multifactorial with important facets defined by many investigators in the field. We hypothesize these other factors might predispose or strengthen the effect of p-EDs to explain why all patients that are exposed to p-EDs do not develop HS. We agree with the reviewer that some of these co-exposure factors are likely genetic and environmental.

As the authors note starting on line 328, this study raises important questions for future study. The authors stated on l 324 that they have identified the molecular mechanism of p-EDS causing HS-like phenotypes, in vitro. I do not believe that this has been demonstrated adequately. There is a correlation between p-EDs levels and HS and this continues in dermal fibroblasts cultured ex vivo, but the manuscript falls short of demonstrating causation. Perhaps a better presentation of critical evidence would be more persuasive but the manuscript in its current form does not.

We show three important findings in this manuscript in the context of the appreciated role of GS in monogenic HS: 1) in MSI of HS skin there are greater p-EDs adducts 2) in HS fibroblasts there is lower NCSTN and 3) p-EDS can inhibit NCSTN. These combine to suggest causation, but we agree with reviewer the only definitive proof would be to cure HS by selectively removing all p-EDs from the system. We have included this important caveat in our discussion and also eliminated any definitive claims of causation in the manuscript.

Important issues:

Overall, I find the data to be extensive and of apparent high quality. However, one is left with the impression that there is a tremendous amount of data that is hard to interpret because it is crammed into a very small space. This is compounded by inadequate discussion of what these extensive data show. I believe that the authors should strive to make the data depicted readily accessible to the readership of Nature Communications and keep only the essential panels from each figure in the main results and move the remainder to Extended Data.

Thank you for the feedback, we strengthened the discussion to be a more expansive and thoughtful dissection of the results. Given the requests from the other reviewers we have not moved a large amount of data into the extended data.

Another important issue with respect to the figures is the extremely small size of the panels in each. I do not understand why the authors did not make the figures themselves full page with the legends on an additional page for the purposes of review. This makes it extremely difficult to interpret the figures. As noted above, it could be beneficial to reduce the number of panels in each figure so that they can be presented at a resolution that is accessible to the reader.

We thank the reviewer for informing us of this issue; we have uploaded larger images of higher resolution to ensure readability and easier viewing.

Figure 2f is particularly important to the author's model since this reflects siRNA knockdown of Nicastrin and its effect on the expression of various genes that are presumably changed in HS. While I appreciate the compact nature of this figure and how clearly the differences between fibroblasts and keratinocytes are shown, the figure really fails to make the point that particular genes whose expression is altered in HS are similarly affected in the siRNA-mediated knockdown. This would be much better shown in a table that relates which genes are the most affected in HS and their corresponding change in the siRNA knockdown.

While we agree with the reviewer that siNCSTN is an important experiment to show the importance of NCSTN in fibroblast function, we do not see a strong parallel between a simple NCSTN knockdown and published HS transcriptomics. This is likely because a loss of NCSTN protein is only one change that occurs in HS among likely dozens or even hundreds of important alterations. Related to this, p-ED exposure is likely a better overall representation of HS biology, as it decreases NCSTN protein and almost certainly has many other effects we have yet to fully characterize. This is apparent in our Fig. 4m, which shows the whole cell phospho-proteomics of HSvsNAX is strongly correlated to p-EDs vs Vehicle treatments. Regardless, we have cross referenced fibroblast gene expression changes in our scRNAseq meta-analysis and our fibroblast bulk RNAseq to identify genes that are differentially expressed in the same direction with strong adjusted p values (see below, now included in **Supplemental Figure 3h**).

Gene	scRNAseq		Bulk RNAseq	
	avg_log2FC	p_val_adj	log2FoldChange	p_val_adj
INHBA	1.745797656	3.52E-80	1.761306682	4.00E-88
WISP2	-2.402565857	3.33E-97	-1.114240464	0.00364048
SEMA3B	1.312189549	5.20E-63	2.196664031	7.44E-36
PTGIS	-1.039756868	5.03E-30	-1.361828624	4.80E-10
MMP9	0.998948453	7.25E-45	1.764644034	1.93E-86

What is being merged in Figure 4k? This is not noted in the legend or in the figure. The lettering throughout the figure panels is too small in many cases to identify what is being depicted.

We thank the reviewer for their feedback. We have increased the lettering size in our figures to increase the readability of the depiction. In the indicated case of Fig. 4k, the merge is NCSTN (red), PDGFRa (green), and DAPI (blue). We have updated the panel to accurately reflect what is included in the merge.

Extended Data figure 5i is said to show a comparison of bulk RNAseq from normal dermal fibroblasts treated with the p-ED mixture vs. normal dermal fibroblasts treated with a gamma secretase inhibitor. As noted above, the panel is uninterpretable as presented, making it difficult to ascertain the validity of the authors' assertions.

We thank the reviewer for letting us know this issue; we have uploaded a larger image of higher resolution to ensure readability to the audience.

Other comments to be addressed:

l 39-41 – It is unclear why the authors switched terminology to plasticizers here since bisphenols are not usually considered to be plasticizers

Thank you for the correction, we have corrected this to use “p-EDs”

The Western Blots in Fig 1g showing are not terribly convincing with respect to the effects of TNF on the cultured fibroblasts.

Thank you for this comment. We have evaluate NCSTN both by western blot and IF with overlapping results to ensure rigor. Because we normalize to our loading protein control, the westerns are not as convincing as the quantitation.

The immunofluorescence panels are very difficult to evaluate. One hopes that these will be large enough in the final version of the paper for the reader to distinguish the features being discussed. I cannot and can only take the descriptions by the authors at face value.

Thank you for the feedback; we have uploaded larger versions of the figures for better evaluation upon resubmission

Perhaps the quantitation of the panels could be shown in the main figures and the immunofluorescence shown with greater magnification and accessibility similar to Extended Data Figure 2?

We have uploaded larger versions of the figures for better evaluation upon resubmission and hope this will rectify the problem.

Reviewer #1 (Remarks to the Author):

Dear Editor,

The Authors have responded to most of the main concerns raised in the first review. The manuscript has been greatly improved by experiments added by the Authors and a better description of patients under study

We are very grateful to partner with the reviewer to help strengthen the manuscript and further the field's efforts to help our HS patients.

I still have some questions about the author's replies.

I understand the Authors have restricted information on the genetics of HS in the patients analysed, as family data are unavailable for most patients -still, adding available data has improved the characterisation of the cohort under study.

I have an issue regarding the choice of healthy controls and patients in this study.

HS affects mainly African American women and 90% of the patients are actually African American, why the Authors used other populations (mainly Caucasians and Asians) as controls?

We agree African American controls would be ideal. Our study participants were patients without HS who agreed to donation biopsies, which despite our efforts did not include as many African American patients as we wished. We agree this is a limitation of the study, and we have mentioned this in the discussion to raise attention of the readers to this important point.

Most of the healthy controls are overweight – do they consume ultra-processed foods as well, or do they have a completely different diet from HS patients?

Do the Authors have any information regarding diet preferences in the two cohorts?

While we do not have data specifically on the preferences of patients in our cohort, a recent publication from our group explores diet preferences between HS and control patients and found the most significant difference in diets was single use plastic water bottles, known to be high in p-EDs and microplastics (PMID: 39481530). While we did make an attempt to ensure that the healthy controls were also overweight to match this feature to our HS patients, we did not find it feasible to try to match dietary habits given the large variety and lack of simple, practical, and well-established quantitative indices of nutritional habits. Therefore, it is a limitation of our study that we do not have full nutritional surveys of all the participants. The reviewer brings up a vital area of future study that we have also described in the discussion to help mobilize the field towards the very important questions the reviewer raises.

How do the Authors explain that TNFa plays a major role in CXCL8's increased expression by fibroblasts in their patients, and yet many of them still failed to respond to Humira?

Is it worth to add few lines in the discussion section

We agree this is an interesting point. Most patients *did* at least have a partial if time-limited and temporary response to Humira in the early stages of their treatment and disease. But the

reviewer is correct that many of our subjects eventually failed Humira. This raises an important point that we agree needs emphasis—our study is likely just one of many parallel biologic pathways where HS disease is promoted. Besides pED exposure there are likely co-exposures or genetic factors that modulate disease. We have emphasized this in the discussion to try to promote more work by other investigators in the field to help better define this disease so that together we may discover new treatments for our patients through studying the multifactorial pathogenesis of HS.

For most of the experiments 8 HS were used and there are 12 patients enrolled in this study. May the Authors point which patients were actually analysed?

All individual patients are listed for each specific experiment either in the main figures or in supplementary figures, and is additionally broken out from pooled to individual data in the supplementary data. Some patient samples did not have sufficient tissue for all experiments, and were used as available.

Figure 1 of the paper and rebuttal letter: Supplemental Fig. 7

I am concerned about Western blot and IF analyses on NCSTN expression. The antibody used in this study can detect both the mature and immature forms of the NCSTN protein.

Did the Authors analyze the mature and immature forms of the protein or they focus on just one form?

Actually, most of the figures throughout the manuscript (especially siRNA experiments) showed just one main band for the protein.

We attempted several different NCSTN antibodies before settling on the use of MA529438 and PA517735 (also attempted PA1-758, 14071-1-AP, B-3 Santa Cruz, MA5-40999, among others) and but were unable to identify an antibody capable of distinguishing mature and immature NCSTN reliably and repeatably. That being said, we believe the immature (non-glycosylated) form of NCSTN is typically visible in the WBs, just very faintly, as in Figure 4i; note the faint second band directly below the larger NCSTN band ~10-20 kDa away. Because we cannot reliably distinguish these two forms via WB, we chose to include all NCSTN protein that appeared on our WBs from our NCSTN antibody in our analyses. See below as an example from siRNA experiment in Fig 3a. We thought this would be the least biased, most complete way to represent NCSTN protein in these experiments.

Original Blot

Possible Mature/Immature NCSTN bands

How bands were analyzed

*IF experiments cannot differentiate between the 2 forms, but WB gels should be shown to help readers understand which form was visible and analysed by the Authors.

See above. All original western blots will be uploaded to the journal for access to the community.

For the supplemental Figure 7 both mature and immature forms of the protein should be analysed to understand the role of a defect in the maturation process of the protein.

In these experiments, despite using the same antibody, our “immature” NCSTN band was not visible, even when the blot contrast was increased significantly (see below). We analyzed them the same way as shown previously, to ensure we included all NCSTN protein that may be in the lane.

Because of the infrequent ability to detect “immature” forms of NCSTN it is unfortunately not feasible to more accurately quantify this phenomenon. Given the reviewer’s concerns, we have also added in the discussion the need to carefully dissect NCSTN mRNA post-translational maturation and modifications as an important area for the field to address.

Still, it is already known that a lack of a single subunit of the gamma-secretase complex may increase the degradation of g-secretase subunits.

Given the new results shown by the Authors that stated that : “ GS subunit mRNA are similarly affected by p-ED exposure in the presence of TNF α , so the effect is at least partially transcriptionally regulated and interestingly inhibiting multiple GS subunits” how can they be sure that p-ED exposure affects the transcription of an unspecified endonuclease able to specifically degrade NCSTN and do not affect the expression of other g-secretase subunits that cause an increased instability of mature NCSTN, thus reducing the levels of the total protein?

The reviewer brings up an important point that there are many potential explanations for decreased NCSTN with pEDs, including GS subunit co-dependency. We agree the suggestion of an endonuclease is extremely hypothetical. We have emphasized the hypothetical nature of this conjecture, and also the likelihood of other more likely explanations including GS subunit co-dependency.

The Authors never provided an experimental link between decreased NCSTN expression and pNF-kB nuclear translocation. As the authors focused only on NCSTN expression, it is unclear whether this could be related to the γ -secretase complex activity or NCSTN activity independent of other γ -secretase subunits.

We believe the pNF-kB activity is elevated due to γ -secretase activity due to the loss of NCSTN. But we very much agree with the reviewer that we cannot rule out a role of NCSTN independent of the GS complex that might modulate nuclear pNF-kb. We have inserted into the discussion the important point that we cannot rule out GS-independent functions of NCSTN in our manuscript.

It is unclear whether the Authors support the hypothesis that HS is primarily an autoinflammatory keratinisation disease, where the first insult is an auto-inflammatory response or is a disorder driven by the occlusion of the hair follicle.

In any case, where will the authors place an elevated consumption of p-ED in the pathophysiology of the disease?

Is an elevated consumption of processed food associated with the disease severity or may it cause the disease itself?

Can you please clarify this point in the discussion section?

The reviewer raises the important need to better contextualize our work in the framework of the entire field. Fundamentally we believe that our results suggest a role of p-EDs in worsening HS disease, but do not address the larger questions raised by the reviewer. Most importantly, our results are likely one facet of a multifactorial disease. We have emphasized these points in the discussion.

Beyond this, we can speculate about where pEDs might fit in the pathogenesis. Given the work by others in the field it seems likely that HS is both an autoinflammatory disease and a hair follicular occlusion disease. It is not absolutely clear which, if either, occur first, or exactly which may drive the other. Given that HS is an inflammatory disease, and our findings suggest that pEDs induce NCSTN loss to potentiate inflammation, we advocate that p-EDs likely contribute to disease severity. Hair follicle occlusion may be partially caused by an inflammatory cascade via our hypothesis outlined in this paper or may be an independent process that occurs primarily due to notch signaling deficits in keratinocytes. All these processes are vitally important for disease pathogenesis and are important areas of future study.

I want to thank the Authors for addressing the remaining points raised in the first review of this manuscript.

Reviewer #2 (Remarks to the Author):

The authors replied satisfactorily to this reviewer by supplementing additional experimental

data.

Reviewer #4 (Remarks to the Author):

The authors have responded to my critiques in a satisfactory way. I am still not convinced that they have established a causal relationship between plasma bisphenols and HS, but the data presented here are a valuable addition to the literature.